



# Cryptic role of tetrathionate in the sulfur cycle: A study from Arabian Sea oxygen minimum zone sediments

Subhrangshu Mandal[1], Sabyasachi Bhattacharya[1], Chayan Roy[1], Moidu Jameela Rameez[1],
Jagannath Sarkar[1], Svetlana Fernandes[2], Tarunendu Mapder[3], Aditya Peketi[2], Aninda
Mazumdar[2,*] and Wriddhiman Ghosh[1,*]

[1] Department of Microbiology, Bose Institute, P-1/12 CIT Scheme VIIM, Kolkata 700054, India.
[2] CSIR-National Institute of Oceanography, Dona Paula, Goa 403004, India.
[3] ARC CoE for Mathematical and Statistical Frontiers, School of Mathematical Sciences,
Queensland University of Technology, Brisbane, QLD 4000, Australia.

**\* Correspondence emails:**     wriman@jcbose.ac.in / maninda@nio.org
**Running Title:**             Tetrathionate metabolism in marine sediments

**KEYWORDS:** sulfur cycle, tetrathionate, marine oxygen minimum zone, sediment biogeochemistry

---

**ABSTRACT** To explore the potential role of tetrathionate in the sulfur cycle of marine sediments, the population ecology of tetrathionate-forming, oxidizing, and respiring microorganisms was revealed at 15-30 cm resolution along two, ~3-m-long, cores collected from 530- and 580-mbsl water-depths of Arabian Sea, off India's west coast, within the oxygen minimum zone (OMZ). Metagenome analysis along the two sediment-cores revealed widespread occurrence of the structural genes that govern these metabolisms; high diversity and relative-abundance was also detected for the bacteria known to render these processes. Slurry-incubation of the sediment-samples, pure-culture isolation, and metatranscriptome analysis, corroborated the *in situ* functionality of all the three metabolic-types. Geochemical analyses revealed thiosulfate (0-11.1 μM), pyrite (0.05-1.09 wt %), iron (9232-17234 ppm) and manganese (71-172 ppm) along the two sediment-cores. Pyrites (via abiotic reaction with $MnO_2$) and thiosulfate (via oxidation by chemolithotrophic bacteria prevalent *in situ*) are apparently the main sources of tetrathionate in this ecosystem. Tetrathionate, in turn, can be either converted to sulfate (via oxidation by the chemolithotrophs present) or reduced back to thiosulfate (via respiration by native bacteria); 0-2.01 mM sulfide present in the sediment-cores may also reduce tetrathionate abiotically to thiosulfate and elemental sulfur. Notably tetrathionate was not detected *in situ* - high microbiological and geochemical reactivity of this polythionate is apparently instrumental in the cryptic nature of its potential role as a central sulfur cycle intermediate. Biogeochemical roles of this polythionate, albeit revealed here in the context of OMZ sediments, may well extend to the sulfur cycles of other geomicrobiologically-distinct marine sediment horizons.

---

## 1 Introduction

Phylogenetically diverse microorganisms oxidize and/or reduce different sulfur species to meet their bioenergetic requirements, and in doing so play profound roles in running biogeochemical sulfur cycles in nature (Baumgartner et al., 2006; Ghosh and Dam, 2009). Within the marine



realm, microbial processes of sulfur cycle are central to benthic biogeochemistry as they are linked to the *in situ* transformations, sequestrations, and fluxes of carbon and iron. So there have been extensive studies of benthic/sedimentary sulfur cycle across the global ocean (Jørgensen,
1990; Jørgensen and Bak 1991; Rudnicki et al., 2001; Tostevin et al., 2014), and focus has typically been on geomicrobial transformations of the two end-members sulfate and sulfide (Holmkvist et al., 2011; Jørgensen et al., 2019), and also thiosulfate which constitutes a key junction in the network of sulfur species transformation in marine sediments (Jørgensen, 1990; Jørgensen and Bak 1991 Thamdrup et al., 1994). In this context, however, tetrathionate or other
polythionates are seldom appreciated for their potential role(s) in marine sedimentary sulfur cycle, presumably because these sulfur species are not abundant in these environments. Overall paucity of polythionates in any environment is largely attributable to their high reactivity under biotic (Kanao et al., 2007; Ghosh and Dam, 2009; Boden et al., 2010; Pyne et al., 2017, 2018) as well as abiotic (Schippers et al., 1999; Schippers and Jørgensen, 2001) conditions. Cryptic
nature of these sulfur species notwithstanding, several bacteria capable of producing and/or utilizing tetrathionate for bioenergetic purposes have been isolated from different terrestrial and aquatic (both fresh-water and marine) habitats (Kaprálek, 1972; Oltmann and Stouthamer, 1975; Barrett and Clark, 1987; Sass et al., 1992; Price-Carter et al., 2001; Sorokin, 2003; Ghosh et al., 2005; Ghosh and Roy, 2006, 2007; Ghosh and Dam, 2009). Here we investigate the potential
involvement of tetrathionate in the sulfur cycle of a marine sediment horizon located within the perennial oxygen minimum zone (OMZ) of the eastern Arabian Sea, off the west coast of India. Community structures and functions of tetrathionate-forming, oxidizing, and respiring, microorganisms were revealed (via metagenomics, metatranscriptomics and pure-culture isolation) along two ~3-m-long sediment cores (Fernandes et al., 2018) collected from 530 and
580 meters below the sea level (mbsl); pore-water/solid-phase chemistry was characterized using a range of analytical techniques; the microbial ecology was then considered in the context of the *in situ* geochemistry, and implications were inferred for the sedimentary sulfur cycle.

## 2 Materials and methods

### 2.1 Study site and, sample collection and storage

During the comprehensive exploration of the sediment biogeochemistry of eastern Arabian Sea OMZ (ASOMZ) on-board RV Sindhu Sankalp (SSK42), the two gravity cores SSK42/5 and SSK42/6, on which the present study is based, were collected from 580 mbsl (16°49.88' N, 71°58.55' E) and 530 mbsl (16°50.03' N, 71°59.50' E) water-depths respectively (Fernandes et
al., 2018). These two sediment-cores, collected from the approximate center of the vertical expanse of the ASOMZ off the west coast of India, were both ~3 m long; onboard-sampling of the cores was carried out at 15 to 30 cm intervals. The gravity cores SSK42/5 and 6 were sampled,





as described previously, under constant shower of high-purity $N_2$ to avoid aerial oxidation of the $H_2S$ and $Fe^{2+}$ potentially present in the sediments (Fernandes et al., 2018). Adequate measures

were taken to avoid post-sampling contamination of the native microbial communities and physicochemical alteration of the geochemical properties of the sediments. Immediately after the longitudinal halves of the PVC core-liners were cut open, top one cm of the exposed surfaces of the individual cores were scrapped off using sterile scalpels to eliminate potential contaminations from the core-liners' inner-surfaces and/or sea-waters through which the cores had passed

(Fernandes et al., 2018). Sample bottles were flushed with high-purity $N_2$ before they were screw-capped, sealed with Parafilm (Bemis Company Inc., Neenah, WI, USA), and refrigerated. For every sediment-sample, two fractions designated for duplicate metagenome analysis were put into -20°C freezers, while one fraction each for chemistry and culture-based microbiological studies were kept at 4°C. These temperatures were all the time maintained during transfer of the

samples to the institutes, and also over their long-term preservation in the laboratories.

### 2.2 Geochemical analyses

Concentration of dissolved thiosulfate in the pore-waters of the sediment-samples was determined by anion chromatography using an Eco IC (Metrohm AG, Herisau, Switzerland)

equipped with a conductivity detector (Metrohm, IC detector 1.850.9010). Chemical suppression was used for this purpose, while separation was carried using a Metrosep A Supp5 - 250/4.0 (6.1006.530) anion exchange column (Metrohm AG); a mixed solution of 1.0 mM sodium hydrogen carbonate and 3.2 mM sodium carbonate was used as the eluent; 100 mM sulfuric acid was used as the regenerant; flow rate was 0.7 mL min $^{-1}$, and injection volume 100 µL. Prior to

analysis, pore-water samples were diluted 1000-fold with de-ionized water (Siemens, <0.06 µS) and passed through 0.22 µm hydrophilic polyvinylidene fluoride membranes (Merck Life Science Private Limited, Bengaluru, India). Sigma-Aldrich (St. Louis, MO, USA) standard chemicals were used to prepare the calibration curve for quantification. Overall sample reproducibility was ±0.2 ppm.

110          Presence of iron and manganese in the bulk sediment-samples was determined by X-ray fluorescence (XRF) spectrometry using an ARTAX 200 portable µ-XRF spectrometer (Bruker Scientific Instruments, Billerica, MA, USA) having an Mo Tube X-ray source and Si-drift detector. Voltage and current used to excite the secondary fluorescence X-rays were 50 kV and 698 µA, respectively; other operational parameters were as follows: count rate, 1877 cps; optic used to

collect the emitted secondary X-rays, Collimator 0.650; live time, 60 s; dead time, 0.4 %; atmosphere in which samples were analyzed, air. Subsequently, the two elements were quantified in the bulk sediment-samples using an Agilent 240 AA flame-atomic absorption spectrometer (FAAS) (Agilent Technologies, Santa Clara, CA, USA). Sediment-samples were



prepared following the United States Environmental Protection Agency's prescribed method for
acid digestion of sediments, sludge and soils (protocol number 3050B), where 0.5 g sediment-
sample was taken in a beaker containing 50 mL distilled water plus 10 mL each of concentrated
nitric acid and hydrochloric acid; the beaker was covered with watch glass and placed on a hot
plate; digestion was continued until silica alone remained as the precipitate; finally the sample
was cooled and filtered; volume was adjusted to 100 mL with distilled water, and then that
solution was analyzed by FAAS. The FAAS instrument was operated with the wavelengths 248.3
and 279.5 nm for iron and manganese respectively; lamp current for both elements was 5 mA;
fuel gas used was acetylene, while support gas was air; oxidizing condition was used for flame
stoichiometry. For preparing standard curves ICP multi-element standard solution IV (Merck
KGaA, Darmstadt, Germany) was used at a concentration of 1000 mg mL$^{-1}$.To estimate pyrite,
acid volatile sulfur (AVS) and chromium reducible sulfur (CRS) were extracted sequentially with
cold 6N HCl and boiling 1M $CrCl_2$ in 6N HCl solution in an oxygen-free environment, as described
previously (Mazumdar et al., 2012). The evolved $H_2S$ was precipitated as $Ag_2S$ for quantification
of CRS and AVS. Whereas AVS was not detected in the samples from SSK42/5 and 6, the CRS
values represented the S contributed from the $FeS_2$ (pyrite) present.


**2.3 Extraction of total DNA/RNA from sediment-samples/pure-culture isolates**

Total community DNA was extracted from the sediment-samples using PowerSoil DNA Isolation
Kit (MoBio, Carlsbad, CA, USA), as per the manufacturer's protocol. Microgram-level of DNA was
obtained from each batch of preparatory reaction that started with 0.5 g sediment-sample.
Genomic DNA of pure culture isolates was extracted using HiPurA Bacterial Genomic DNA
Purification Kit (Himedia Laboratories, Mumbai, India), following manufacturer's instructions.
Quality of metagenomic/genomic DNA samples was checked by electrophoresis and considered
to be of high quality when no degradation signs were apparent. DNA quantity was determined
using Qubit dsDNA HS Assay Kit (Thermo Fisher Scientific, Waltham, MA, USA).

145       Total community RNA was extracted from the 275 cmbsf sediment-sample of SSK42/6
using the RNA PowerSoil Total RNA Isolation Kit (MoBio), as per manufacturer's protocol.
Nanogram-level total RNA was obtained after pooling up the products of 15 individual
preparatory reactions, each carried out using 2 g sediment-sample. All the individual RNA
preparations were subjected to DNase digestion by RNase free DNase I (Thermo Fisher
Scientific) and purified using RNeasy MinElute Cleanup Kit (Qiagen, Hilden, Germany); their
concentrations were measured using Quant-iT RiboGreen RNA Assay Kit (Thermo Fisher
Scientific). Integrity of RNA (RIN) within the individual preparations was determined on a
TapeStation RNA ScreenTape electrophoretic system (Agilent Technologies) and only high-



quality preparations having RIN value >7.0 were added to the RNA pool that were subsequently

used for sequencing library construction.

## 2.4 Metagenome (total community DNA) sequencing

The duplicate set of total community DNA (metagenomes) isolated for each sediment-depth explored along SSK42/5 and 6 were shotgun sequenced individually on an Ion Proton

sequencing platform (Thermo Fisher Scientific) using 200 nucleotide read-chemistry, as described previously (Ghosh et al., 2015). Complete lists of sedimentary communities investigated along SSK42/5 and 6 are given in Tables S1 and S2 respectively.

1 µg DNA from each sediment-sample was taken for deep shotgun sequencing by the Ion Proton platform using 200 bp read chemistry on a PI V2 Chip. Sequencing libraries were

constructed using Ion Plus Fragment Library Kit (Thermo Fisher Scientific), following the manufacturer's Ion Plus gDNA Library Preparation User Guide. The Proton library was generated using 1 µg of genomic DNA which was fragmented to approximately 200 base pairs by the Covaris S2 system (Covaris, Inc., USA) and purified with 1.8X Agencourt Ampure XP Beads (Beckman Coulter, USA). Fragmentation was followed by end-repair, blunt-end ligation of the Ion

Xpress Barcode and Ion P1 adaptors, and nick translation.

Post-ligation, size selection was done using E-Gel Size-Select 2% Agarose gels (Thermo Fisher Scientific) with 300 bp target size. Final PCR was performed using platinum PCR SuperMix High Fidelity and Library Amplification Primer Mix (Thermo Fisher Scientific), for 5 cycles of amplification. The resulting library was purified using 1.2X AMPure XP reagent

(Beckman Coulter) and the concentration determined with Qubit dsDNA HS Assay Kit (Thermo Fisher Scientific); size distribution was done with Agilent 2100 Bioanalyzer high-sensitivity DNA kit (Agilent Technologies). Libraries were pooled in equimolar concentrations and used for template preparation.

Library templates for sequencing were prepared using OneTouch 2 protocols and

reagents (Thermo Fisher Scientific). Library fragments were clonally amplified onto ion sphere particles (ISPs) through emulsion PCR and then enriched for template-positive ISPs. Proton emulsion PCR reactions utilized the Ion PI Template OT2 200 Kit v3 (Thermo Fisher Scientific). Following recovery, enrichment was completed by selectively binding the ISPs containing amplified library fragments to streptavidin coated magnetic beads, removing empty ISPs through

washing steps, and denaturing the library strands to allow for collection of the template-positive ISPs. For all reactions, these steps were accomplished using the ES module of the Ion OneTouch 2. The selected ISPs were loaded on PI V2 Chip and sequenced with the Ion PI 200 Sequencing Kit (Thermo Fisher Scientific) using the 500 flow (125 cycle) run format.





### 2.5 Metatranscriptome (total community mRNA) sequencing

The pooled-up total RNA preparations were selectively converted to a library of template molecules using TruSeq Stranded mRNA and Total RNA kit (Illumina Inc., San Diego, CA, USA). Depletion of rRNAs was carried out using the Ribo-Zero Gold system (Illumina Inc.), which is an integral part of the kit used for preparing the library. The rRNA-depleted RNA pool, which was

expected to contain only the total mRNA, was fragmented into small pieces using divalent cations under elevated temperature. The cleaved RNA fragments were copied into first strand cDNAs using reverse transcriptase and random primers. This was followed by second strand cDNA synthesis using DNA Polymerase I and RNase H. cDNA fragments were then subjected to end-repair, addition of single 'A' bases, adaptor ligation, purification and enrichment with PCR to

create the final library, which was sequenced on a HiSeq4000 platform (Illumina Inc.) using paired end, 2 × 150 nucleotide, sequencing by synthesis read-chemistry with dual indexing workflows. Furthermore, in order to extract and eliminate any rRNA read that may have been there in the raw metatranscriptomic sequence dataset, the 26,579,343 read-pairs available in all were mapped onto SILVA large subunit as well as small subunit rRNA gene sequence database

(Quast et al., 2012), using the short read aligner Bowtie2 v.2.3.4.3 (Langmead and Salzberg, 2012) in default local (sensitive) alignment mode. This identified ~0.3% reads as ascribable to rRNAs, thereby leaving 26,496,769 read-pairs in the final dataset used for downstream analyses.

### 2.6 *De novo* assembly and annotation of metagenomes/metatranscriptome

All metagenomic sequence datasets were quality-filtered with Phred score cut-off 20 using Fastx_toolkit 0.0.13.2 (http://hannonlab.cshl.edu/fastx_toolkit/download.html). High quality reads from the duplicate metagenomic sequence datasets available for each sediment-community were co-assembled using Megahit v1.2.x (Li et al., 2015) with the kmers 21, 29, 39, 59, 79, 99, 119 and 141 for a minimum contig-length of 100 bp; each assembly obtained in this way was quality-

checked using MetaQUAST (Mikheenko et al., 2015), and >100-bp-long contigs were searched for ORFs/genes encoding >30-amino-acids-long peptides using MetaGeneMark (Zhu et al., 2010).

The rRNA-sequence-free metatranscriptomic dataset was assembled using the python script rnaspades.py, available within SPAdes 3.13.0 (Nurk et al., 2013), with default parameters.

ORFs/genes encoding continuous stretches of minimum 30 amino acids were predicted in contigs longer than 100 bp using Prodigal v2.6.3 (Hyatt et al., 2010).

Gene-catalogs obtained from the individual metagenomes/metatranscriptome were functionally annotated by searching against EggNOG v5.0 database (http://eggnog5.embl.de/download/eggnog_5.0/) with EggNOG-mapper (Huerta-Cepas et al.,

2016) (http://beta-eggnogdb.embl.de/#/app/emapper) using HMMER algorithms. Enzymes



involved in tetrathionate formation, tetrathionate oxidation, tetrathionate respiration, and manganese oxidation were screened manually on the basis of their KEGG Orthology numbers (Kanehisa et al., 2016).

**2.7 Direct taxonomic/functional annotation of raw metagenomic reads**

Raw (unassembled) reads contained in the duplicate metagenomic sequence datasets obtained for each sediment-community were directly annotated taxonomically by separately searching them against the non-redundant (*nr*) protein sequence database of National Center for Biotechnology Information (NCBI), Bethesda, MD, USA, using the Organism Abundance tool of

MG-RAST 3.6 (Meyer et al., 2008). The two independent values obtained in this way for the relative abundances of taxa within a community (namely, genera of tetrathionate-formers, oxidizers and reducers, and genera of manganese-depositors and oxidizers) were averaged and used for comparisons between communities. In these analyses, percentage allocation of reads over various taxa was taken as a direct measure of the prevalence of those taxa within the

community (Ghosh et al., 2015). Within MG-RAST, sequences were trimmed to contain no more than 5 successive bases with phred score <15. To classify reads using Organism Abundance tool, Best Hit Classification algorithm was followed [BlastX search with minimum 45 nucleotides (15 amino acids) alignment and ≥60% identity, and maximum $e$-value allowed $1e^{-5}$].

**2.8 Slurry culture experiments**

Abilities of the individual sediment-samples to oxidize thiosulfate, and oxidize/reduce tetrathionate, were tested via aerobic slurry incubation experiments conducted at 15°C on a rotary shaker (150 rpm). For each experiment, 0.5 gm sediment was suspended in 5 mL ASWT or ASWTr broth medium and incubated for 30 days. ASWT or ASWTr medium (pH 7.5) contained

artificial sea water (ASW) supplemented with $Na_2S_2O_3.5H_2O$ (10 mM) or $K_2S_4O_6$ (5mM), added separately after filter-sterilization (Alam et al., 2013). ASW contained the following $L^{-1}$ distilled water: 25.1 g NaCl, 1 g $(NH_4)_2SO_4$, 1.5 g $MgSO_4$, $7H_2O$, 0.3 g $CaCl_2.2H_2O$, 0.2 g $NaHCO_3$, 2.4 g Tris, 1 mL trace element solution and 0.5 g $K_2HPO_4$ (added after autoclaving separately). 1 L trace element solution (pH 6.0), in turn, contained 50 g EDTA, 22 g $ZnSO_4.7H_2O$, 5.06 g $MnCl_2$,

4.99 g $FeSO_4$, 1.1 g $(NH_4)_6 MoO_{26}.4H_2O$, 1.57 g $CuSO_4$ and 1.61 g $CoCl_2.6H_2O$.

To check the tetrathionate-reducing ability of a sample, 2.5 gm sediment was suspended in 45 mL Rappaport Vassiliadis tetrathionate medium (Vassiliadis, 1983) that was contained, and already autoclaved, in a screw-capped bottle. Sediment addition to the medium and subsequent incubation (for 30 days) of the screw-capped bottles were all carried out inside a Whitley H35

Hypoxystation (Don Whitley Scientific, West Yorkshire, UK) preset at 75% humidity, 15°C temperature and 0% partial pressure of $O_2$, using the gas mixture $N_2:H_2:CO_2 = 80:10:10$ (v/v/v).





RVTr medium (pH 5.4) contained the following $L^{-1}$ of distilled water: 4.5 g soya peptone, 8.0 g NaCl, 0.4 g $K_2HPO_4$, 0.6 g $KH_2PO_4$, 29.0 g $MnCl_2$, 0.036 g Malachite green, 10 mM $K_2S_4O_6$ (added separately after filter sterilization) and 0.5 g sodium thioglycolate (to eliminate $O_2$) and 0.1 mg resazurin (to indicate presence of any $O_2$).


Concentration of thiosulfate, tetrathionate and sulfate in the media were measured by iodometric tritation, cyanolysis and gravimetric sulfate precipitation method respectively at different time intervals (Alam et al., 2013). Dissolved sulfides were precipitated from the spent media and subjected to colorimetric measurement based on the principle that N, N-dimethyl-p-phenylenediaminedihydrochloride and $H_2S$ react stoichiometrically in the presence of $FeCl_3$ and HCl to form a blue-colored complex (Cline, 1969).


### 2.9 Enrichment, isolation and characterization of bacterial strains

Isolation of sulfur chemolithotrophs from the 275 cmbsf sediment-sample of SSK42/6 was carried out in mineral salt-thiosulfate-yeast extract (MSTY) and ASWT/ASWTY media. While the ASWTY medium was a yeast extract (500 mg $L^{-1}$) supplemented derivative of ASWT, MSTY contained modified basal and mineral salts (MS) solution supplemented with 20 mM $Na_2S_2O_3.5H_2O$ and 500 mg $L^{-1}$ yeast extract (pH 7.0). MS, in turn, contained the following $L^{-1}$ distilled water: 1 g $NH_4Cl$, 4 g $K_2HPO_4$, 1.5 g $KH_2PO_4$, 0.5 g $MgSO_4.7H_2O$ and 5.0 mL trace metals solution (Vishniac and Santer, 1957). Three portions of the 275-cmbsf sediment-sample of SSK42/6 were added (5% w/v) individually to MSTY, ASWT and ASWTY and broths, and incubated aerobically at 15°C until phenol red indicator present in the media turned yellow (apparently due to production of sulfuric acid from thiosulfate). Post yellowing, individual enrichment slurries were kept undisturbed for 1 h to allow sediment particles to settle down; 10 mL cell suspension from each flask was then centrifuged at 6000 *g* for 10 min and the pellet re-suspended in 1 mL of the corresponding medium, serially diluted, and spread onto agar plates of the corresponding medium, and incubated at 15°C. Morphologically distinct colonies were picked up and dilution streaked till all colonies in individual plates looked similar; representative colonies from individual pure-plates were taken as strains and maintained in their respective isolation-media. Only *Methylophaga*, though isolated in ASWT, was maintained in ASW supplemented with 0.3% (v/v) methanol (i.e. ASWM) because its growth in ASWT waned after six straight sub-cultures.

Chemolithotrophic capabilities of the new isolates were tested in MSTY, MSTrY, ASWT, ASWTY, ASWTM, ASWTr. MSTrY contained MS solution supplemented with 10 mM $K_2S_4O_6$ and 500 mg $L^{-1}$ yeast extract; ASWTM contained ASW supplemented with 10 mM $Na_2S_2O_3.5H_2O$ and 0.3% (v/v) methanol; ASWTr contained ASW supplemented with 10 mM $K_2S_4O_6$. Concentrations of dissolved thiosulfate, tetrathionate and sulfate in the spent media were measured as already described above.



Tetrathionate-reducing bacterial strains were isolated from the 275 cmbsf sediment-sample of SSK42/6 in RVTr medium (Vassiliadis, 1983) under strictly anaerobic condition. 2.5 g

sediment-sample was added to 45 mL RVTr broth that was contained, and already autoclaved, in a screw-capped bottle. Sediment addition to the medium, and subsequent incubation of the screw-capped bottles at 15°C for one month, were all carried out inside the Whitley H35 Hypoxystation preset to zero $O_2$ as stated above. After one month, still inside the Hypoxystation, 1 mL of the sediment- RVTr slurry was serially diluted and spread onto RVTr agar plates and

incubated at 15°C. After growth appeared in the RVTr agar plates, they were taken out, repeatedly dilution streaked on to fresh plates and incubated aerobically, till all colonies in the individual plates looked similar. Representative colonies from the pure-plates were taken as strains and maintained in aerobically in Luria Bertani medium. The pure isolates obtained in this way were classified down to the lowest identifiable taxonomic category, as described above for

the sulfur-oxidizing isolates. Tetrathionate-reducing abilities of the new isolates were tested by growing them for 30 days in RVTr broth, inside the H35 Hypoxystation, as described above. Concentrations of dissolved thiosulfate, tetrathionate and sulfide in the spent RVTr medium were also measured as before.

Genomic DNA was extracted from the isolated strains using HiPurA Bacterial Genomic

DNA Purification Kit (Himedia Laboratories) following manufacturer's protocol. Using their genomic DNA as template, 16S rRNA genes were PCR amplified from the individual bacterial isolates strains using the Bacteria-specific universal primer-pair 27f and 1492r (Gerhardt, 1994). 16S rRNA gene sequences were determined from the PCR products using the same universal primers; according to the manufacturer's instructions for a 3500xL Genetic Analyzer automated

DNA sequencer (Thermo Fisher Scientific). The 16S rRNA gene sequence of each strain was compared against sequences available in the GenBank/EMBL/DDBJ databases, using BLASTN; strains were finally classified down to the lowest identifiable taxonomic category on the basis of their 16S rRNA gene sequence similarities with the closest, validly-published species having standing in nomenclature (http://www.bacterio.net/; see also Euzéby, 1997; Parte, 2013).


## 3 Results and discussion

### 3.1 Tetrathionate-forming, oxidizing, or respiring genes and relevant microorganisms are abundant in the sediment horizon of SSK42/5 and 6

When the metagenomic sequence data obtained for each of the 25 distinct sediment-samples of

SSK42/5 and 6 were assembled and annotated individually, 23 out of the 25 contig-collections obtained were found to contain genes for tetrathionate formation (namely, genes encoding subunits of the thiosulfate dehydrogenases TsdA and DoxDA) (Table S3), while all the 25 contig-collections contained genes for tetrathionate oxidation (namely, thiol esterase *soxB* and sulfur





dehydrogenase *soxC* (Table S4). Furthermore, 23 out of the 25 contig-collections were found to
contain genes for tetrathionate reduction (namely genes encoding subunits of tetrathionate
reductase TtrABC and thiosulfate reductases PhsAB and PsrA) (Table S5). *tsdA* (Denkmann et
al., 2012; Pyne et al., 2018) and *doxDA* (Quatrini et al., 2009) are known to be involved in
thiosulfate to tetrathionate conversion by taxonomically diverse chemolitho/heterotrophic
bacteria. *soxB* and *soxC* are known to be involved in tetrathionate (as well as thiosulfate)
oxidation by taxonomically diverse chemolithotrophic bacteria (Friedrich et al., 2001, 2005; Lahiri
et al., 2006; Alam et al., 2013; Pyne et al., 2018; Mandal et al., unpublished data). In diverse
tetrathionate-respiring species, *ttrABC* reduce tetrathionate to thiosulfate or sulfide (Barrett and
Clark, 1987), while *phsAB* and *psrA* convert thiosulfate to sulfide (Stoffels et al., 2011).

Concurrent with the above findings, direct taxonomic annotation of the raw (unassembled)
metagenomic sequence data (by searching against the *nr* protein sequence database of NCBI)
revealed that considerable proportions of the reads obtained from the individual sediment-depths
of SSK42/5 and 6 were ascribable to bacterial genera whose members are known to render
tetrathionate formation, oxidation or respiration. In that way, 1.3-4.36% and 3-7.8% of
metagenomic reads obtained from the individual sample-sites of SSK42/5 (Fig. 1) and 6 (Fig. 2)
were ascribable to the genera *Pseudomonas* and *Halomonas*, marine strains of which are known
to form tetrathionate from the oxidation of thiosulfate (Tuttle, 1980; Sorokin, 2003). Likewise, 0.1-
1.5 and 0.4-6.4% of metagenomic reads obtained from the individual sample-sites of SSK42/5
(Fig. 1) and 6 (Fig. 2) were ascribable to the genera *Acidithiobacillus*, *Halothiobacillus* and
*Thiomicrospira*, all members of which oxidize tetrathionate chemolithotrophically (Ghosh and
Dam 2009); on the other hand, 0.1-0.3 and 0.2-0.4% of metagenomic reads obtained from the
individual sample-sites of SSK42/5 (Fig. 1) and 6 (Fig. 2) were ascribable to the genera
*Citrobacter*, *Proteus* and *Salmonella*, all members of which respire by reducing tetrathionate to
thiosulfate and/or sulfide (Kaprálek, 1972; Barrett and Clark, 1987; Price-Carter et al., 2001).

**3.2 Synchronized population-fluctuation of different tetrathionate-metabolizing types,
along SSK42/5 and 6**

Analyses based on the direct taxonomic annotation of the unassembled metagenomic data from
discrete sediment-depths of SSK42/5 revealed that the relative abundances of reads ascribed to
the genera of tetrathionate-forming, oxidizing, and respiring bacteria fluctuate synchronously
along this sediment core (Fig. 1). Corroboratively, pair-wise Pearson correlation coefficients (CC)
as well as Spearman rank correlation coefficients (RCC) between the prevalence of the three
metabolic-types are also significantly high in SSK42/5 (Fig. 1; Table S6), which indicate the
existence of strong syntrophic interdependence between the three tetrathionate-metabolizing
types in this sediment horizon. Fluctuations in the prevalence of the three metabolic-types,



however, are less synchronous along SSK42/6 (Fig. 2) and corresponding correlation values are also relatively weaker (Table S7).

Consistent prevalence of reads ascribed to the thiosulfate-to-tetrathionate-converting bacterial genera *Halomonas* and *Pseudomonas* in the metagenomes isolated from the different sample-sites of SSK42/5 and 6 (Fig. 1 and 2) indicated that tetrathionate could be bioavailable in

the chemical milieu of this sediment horizon (notably, pure-culture strains belonging to these two genera were also isolated from the 275 cmbsf sample of SSK42/6; see section 3.3 below, and also Fig. 3). Apart from these two, several such genera were also found to be well represented in the metagenomes of SSK42/5 and 6, some or all members of which are known to produce tetrathionate as a free intermediate during the oxidation thiosulfate to sulfate (Tables S8 and S9).

These organisms, namely *Acidithiobacillus*, *Advenella*, *Halothiobacillus*, *Pusillimonas* and *Thiomicrospira*, can well increase tetrathionate availability in the ASOMZ sediments, even as they themselves are potential users of the tetrathionate (Kelly and Wood, 2000; Sievert et al., 2000; Boden et al., 2017). Tetrathionate can also be formed *in situ*, as an intermediate of sulfate/sulfite reduction, by members of the genera *Desulfovibrio* and *Desulfobulbus* (Sass et al.,

1992), which were detected along both the cores via direct taxonomic annotation of the unassembled metagenomic data (see Tables S8 and S9 for the percentages of metagenomic reads ascribed to these genera in the different sediment-samples).

Tetrathionate can be oxidized *in situ* as a potential energy and electron source by obligately chemolithotrophic genera such as *Acidithiobacillus*, *Halothiobacillus* and

*Thiomicrospira* (Ghosh and Dam 2009) that were detected via direct taxonomic annotation of the unassembled metagenomic data (Fig. 1 and 2) and/or isolated as pure cultures from the 275 cmbsf sample of SSK42/6 (Fig. 3). Whilst all members of these genera are known to oxidize tetrathionate to sulfate (Kelly and Wood, 2000; Sievert et al., 2000; Boden et al., 2017), several such genera were also detected (via direct annotation of metagenomic reads) along SSK42/5

and 6, some chemolithotrophic members of which are known to oxidize tetrathionate to sulfate. These organisms, such as *Advenella*, *Bosea*, *Burkholderia*, *Campylobacter*, *Hydrogenovibrio*, *Pandoraea, Pusillimonas, Pseudaminobacter, Sulfurivirga*, *Thiohalorhabdus,* and *Thiobacillus* (see Tables S10 and S11 for the metagenomic read percentages ascribed to these genera in the different sediment-samples) may contribute to further tetrathionate depletion from the sediments.

Tetrathionate in the ASOMZ sediments can also be utilized as a respiratory substrate by tetrathionate-reducing bacteria such as *Citrobacter*, *Proteus* and *Salmonella,* which were detected by direct annotation of metagenomic reads (Fig. 1 and 2) and all members of which are known to respire tetrathionate (Kaprálek, 1972; Barrett and Clark, 1987; Price-Carter et al., 2001). In addition, strains of *Enterobacter* such as those isolated as pure cultures from 275

cmbsf of SSK42/6 (Fig. 3) can add to the depletion of tetrathionate from the sediments.





Furthermore, several such genera were also detected along SSK42/5 and 6 (via direct annotation of metagenomic reads), some members of which are known to respire tetrathionate in the absence of $O_2$ - these included *Alteromonas, Alcaligenes, Desulfotomaculum, Desulfovibrio, Edwardsiella, Morganella, Pasteurella, Providencia, Serratia* and *Shewanella* (see Tables S12

and S13 for the metagenomic read percentages ascribed to these genera in the different sediment-samples).

### 3.3 The tetrathionate-forming/oxidizing microorganisms of the ASOMZ sediments are alive and active *in situ*

Most of the sulfur chemolithotrophic bacteria known thus far, including those which form tetrathionate from thiosulfate and/or oxidize tetrathionate to sulfate, use molecular oxygen ($O_2$) as the terminal electron acceptor (Ghosh and Dam 2009). So it is apparently peculiar that such microorganisms could be alive and active in the sulfide-rich sediments of SSK42/5 and 6 (Fernandes et al., 2018; also see Table 1) located at the center of the ASOMZ, where $O_2$-

penetration depth of the sediments is generally very low (Breuer et al., 2009). However, in a recent study of sedimentary microbial ecology of SSK42/5 and 6 we have revealed that cryptic $O_2$-sources such as perchlorate respiration enable taxonomically and metabolically diversified communities of aerobic bacteria thrive in this outwardly-anoxic environment (Bhattacharya et al., unpublished data). Consistent to the findings of that paper, present aerobic slurry incubations of

most of the sediment-samples of SSK42/5 and 6 in thiosulfate-containing artificial sea water (ASWT) medium resulted in the formation of tetrathionate and/or sulfate, thereby illustrating the live status of the tetrathionate-forming/oxidizing microflora. In SSK42/5, tetrathionate was the sole end-product of thiosulfate oxidation in slurry incubations of the 0, 15, 90 and 160 cmbsf sediment-samples. These samples converted thiosulfate only up to tetrathionate at a rate of 6.45-

17.72 µmol S day$^{-1}$ g sediment$^{-1}$ respectively (Table S14). In contrast, the 45, 60 and 295 cmbsf samples of SSK42/5 first converted thiosulfate to free and detectable tetrathionate at a rate of 1.11-6.45 µmol S day$^{-1}$ g sediment$^{-1}$; whilst no sulfate was produced during this period of incubation, the accumulated tetrathionate was subsequently converted to sulfate at a rate of 5.86-13.75 µmol S day$^{-1}$ g sediment$^{-1}$ (Table S15). Samples from the rest of the five sediment-

depths explored in SSK42/5 did not metabolize thiosulfate at all. In SSK42/6, tetrathionate was the sole end-product of thiosulfate oxidation in slurry incubations of the 120, 175 and 275 cmbsf sediment-samples. These samples converted thiosulfate only up to tetrathionate at a rate of 17.2-29.71 µmol S day$^{-1}$ g sediment$^{-1}$ (Table S16). In contrast, the 2, 30 and 45 cmbsf samples of SSK42/6 first converted thiosulfate to free and detectable tetrathionate at a rate of 21.05-33.68

µmol S day$^{-1}$ g sediment$^{-1}$; whereas no sulfate was produced during this period of incubation, the accumulated tetrathionate was subsequently converted to sulfate at a rate of 24-54 µmol S day$^{-1}$





g sediment$^{-1}$ (Table S17). Samples from the rest of the seven sediment-depths explored in SSK42/6 did not metabolize thiosulfate at all.

During aerobic slurry incubations in tetrathionate-containing artificial sea water (ASWTr) chemolithotrophic medium, a number of sediment-samples oxidized tetrathionate to sulfate. Of the SSK42/5 samples, those from 0, 15, 45, 90, 120, 160 and 295 cmbsf sediment-depths oxidized tetrathionate at a rate of 2.5-23.5 μmol S day$^{-1}$ g sediment$^{-1}$ (Table S18). Samples from the remaining five sediment-depths explored in SSK42/5 did not oxidize tetrathionate. In SSK42/6, sediment-samples from 2, 30, 45, 60, 75 and 90 cmbsf oxidized tetrathionate at a similar rate of 141 (± 1) μmol S day$^{-1}$ g$^{-1}$ sediment, while the samples from 120, 135 and 175 cmbsf did so at a common rate of 40 (± 1) μmol S day$^{-1}$ g sediment$^{-1}$, and those from 220, 250, 265 and 275 cmbsf at 75 (± 2) μmol S day$^{-1}$ g sediment$^{-1}$.

Results of the slurry culture experiments illustrated that tetrathionate-forming and oxidizing bacteria of SSK42/5 and 6 were alive *in situ*. In order to further verify whether these metabolic-types were functionally (metabolically) active in their native habitat, whole metatranscriptome of the 275 cmbsf sediment-sample of SSK42/6 was sequenced, and the paired end reads assembled into contigs. The gene-catalog obtained via annotation of the assembled contigs were found to encompass homologs of thiosulfate dehydrogenase (*tsdA*), thiol esterase (*soxB*) and sulfur dehydrogenase (*soxC*) (Table S19).

Furthermore, from 275 cmbsf of SSK42/6, 15 such aerobic bacterial strains were isolated in different chemolithoautotophic/mixotrophic thiosulfate-containing media (Table 2) that could form tetrathionate from thiosulfate and/or oxidize tetrathionate to sulfate when grown in corresponding chemolithoautotophic/mixotrophic medium (Fig. 3). 16S rRNA gene sequence-based taxonomic identification of the isolates clustered them under six species-level entities belonging to six distinct genera – the isolates belonging to the genera *Halomonas*, *Methylophaga*, *Pseudomonas* and *Stenotrophomonas* formed tetrathionate from thiosulfate, while those belonging to *Pusillimonas* not only formed tetrathionate from thiosulfate but also oxidized tetrathionate to sulfate; the *Halothiobacillus* isolates did not form tetrathionate and only oxidized the same to sulfate (Table 2). Tetrathionate-forming and/or oxidizing phenotypes of one representative strain each from the six species-level clusters are shown in Fig. 3.

**3.4 Active tetrathionate-reducing microorganisms in ASOMZ sediment**

During anaerobic slurry incubation in heterotrophic Rappaport Vassiliadis medium supplemented with tetrathionate as the sole electron acceptor (RVTr), all the sediment-samples from SSK42/5 and 6 reduced tetrathionate to thiosulfate and/or sulfide at a rate of 0.5-1.5 μmol S day$^{-1}$ g sediment$^{-1}$ (Tables S20 and S21). Notably, no tetrathionate reductase (*ttrABC*) or thiosulfate reductases (*phsAB* or *psrA*) were detected in the gene-catalog obtained via assembly and





annotation of the metatranscriptomic data from 275 cmbsf of SSK42/6; nevertheless, the same catalog did contain many genes having highest sequence identities with functionally diverse genes belonging to the typical tetrathionate-reducer *Salmonella*.


Furthermore, anaerobic enrichment of the 275 cmbsf sediment-samples of SSK42/6, followed by isolation of pure cultures, in RVTr medium yielded four tetrathionate-respiring strains that reduced 30-32 mM S tetrathionate into equivalent amount of thiosulfate over 72 h incubation in RVTr medium (Fig. 3H shows the tetrathionate-reduction kinetics of one representative strain).

16S rRNA gene sequence-based taxonomic identification of the four isolates clustered them under a single species-level entity belonging to *Enterobacter* (Table 2).

**3.5 Thiosulfate and pyrite as key sources of tetrathionate in the OMZ sediments: linking iron and manganese with the sulfur cycle**

In marine sediments, abiotic oxidation of pyrite ($FeS_2$) by $MnO_2$ can lead to the formation of tetrathionate as well as thiosulfate, trithionate, pentathionate and sulfate (Jørgensen and Bak, 1991; Luther, 1991; Berner and Petsch, 1998). The thiosulfate formed in this ways can again be oxidized to tetrathionate either microbially (as mentioned above) or via abiotic reaction with $MnO_2$. Corroborative to these possibilities, results of geochemical analyses (Table 1) revealed up

to 11.1 μM thiosulfate in the pore-waters of all the sulfide-containing sample-sites of SSK42/5 and 6 (tandem absence of sulfide and thiosulfate in the upper 15 cmbsf of SSK42/6 could be due to potentially high rates of chemolithotrophic conversion of sulfide/thiosulfate to sulfate *in situ*). Presence of Fe (9232-17234 ppm), Mn (71-172 ppm) and pyrite (0.05-1.09 wt %) in the solid phase of both the cores (Table 1) lend support to the feasibility of the inorganic sulfide oxidation

pathway envisaged above. Furthermore, when the metagenomic sequence data obtained for each of the 25 distinct sediment-samples of SSK42/5 and 6 were assembled and annotated individually, all 25 contig-collections obtained were found to contain genes for Mn(II) to Mn(IV) oxidation (Table S22). The gene-catalog obtained via assembly and annotation of the metatranscriptomic sequence data from 275 cmbsf of SSK42/6 also encompassed homolog

encoding manganese oxidase (*cotA*) and other accessory proteins involved in Mn(II) to Mn(IV) oxidation (Table S19). Reinforcing the above data, percentage of metagenomic reads ascribed (via direct taxonomic annotation of the raw/unassembled metagenomic reads) to bacteria such as *Aeromonas*, *Citrobacter*, *Enterobacter Gallionella*, *Hyphomicrobium Leptothrix* and *Proteus* that can deposit manganese oxide (MnO) *in situ* (by reducing $Mn^{+4}$ to $Mn^{+2}$ for anaerobic

respiration; see Ghiorse, 1984) was found to be high throughout SSK42/5 and 6 (Fig. 1 and 2). By this type of analysis, again, bacteria which are known to produce $MnO_2$ from MnO [these include *Arthrobacter*, *Oceanospirillum* and *Vibrio* (Tebo et al., 2005; Sujith and Bharathi, 2011)] were also found to be prevalent at all the sample sites (Fig. 1 and 2). In SSK42/5, significantly





positive pair-wise correlation was also found to exist between the relative abundances of
metagenomic reads ascribed to the genera of MnO-depositing bacteria and tetrathionate-forming
/ tetrathionate-oxidizing / tetrathionate-reducing bacteria on one hand and MnO-oxidizing bacteria
and tetrathionate-forming / tetrathionate-oxidizing / tetrathionate-reducing bacteria on the other
(Fig. 1). In SSK42/6, significant positive correlation was observed between MnO-depositing and
tetrathionate-reducing bacteria (Fig 2). These data were reflective of a strong interdependence of
MnO-depositing/MnO-oxidizing bacteria with the different tetrathionate-metabolizing groups.

### 3.6 Concluding remarks

Sulfur cycling is a crucial component of sediment biogeochemistry within the marine realm. Apart
from controlling *in situ* sulfide-sulfate balance, microbe-mediated processes of the sulfur cycle
work in conjunction with those of the carbon cycle to remineralize organic matters sequestered in
the sea-bed, and also influence metal deposition/mobilization. Tetrathionate is seldom
appreciated as a central intermediate of sulfur cycling in marine sediments, even though
thiosulfate is long known to be a central biogeochemical junction of sedimentary sulfur cycling
across the global ocean (Jørgensen, 1990; Jørgensen and Bak 1991; Thamdrup et al., 1994).
Thus far, only one study based on the Baltic Sea sediments has reported microbial production of
tetrathionate and highlighted the role of tetrathionate in the sulfur cycle of Baltic Sea sediments
(Podgorsek and Imhoff, 1999). The present geomicrobiological exploration of the sediments
underlying the approximate-center of the ~200-1200 mbsl vertical-expanse of the Arabian Sea
OMZ revealed tetrathionate as a potent intermediate of the *in situ* sulfur cycle, and identified the
biotic and abiotic mechanisms that are plausibly involved in the formation and transformation of
this polythionate. Albeit the cryptic biogeochemical roles of tetrathionate in the sulfur cycle were
revealed here in the context of an oxygen minimum zone, it is noteworthy that there were no
observable reasons to assume that such processes do not have their equivalents in other
geomicrobiologically-distinct sediment horizons of the marine realm.
Pyrites (via abiotic reaction with $MnO_2$) and thiosulfate (via chemolithotrophic oxidation by
members of the bacterial group designated as A in Fig. 4) are apparently the main sources of
tetrathionate in the sulfidic sediment ecosystem explored. While pyrite can additionally contribute
to the thiosulfate pool, sulfate-/sulfite-reduction (by members of the bacterial group designated as
A' in Fig. 4) can act as an additional source of tetrathionate *in situ*. Tetrathionate formed in this
way can have a number of fates: it can be converted to sulfate (via chemolithotrophic oxidation
by members of the bacterial group designated as B in Fig. 4) or reduced back to thiosulfate (via
respiration by members of the bacterial group designated as C in Fig. 4); copious hydrogen
sulfide present *in situ* (Fernandes et al., 2018) can also reduce tetrathionate abiotically to
thiosulfate and elemental sulfur (Rowe et al., 2015). Tetrathionate, remarkably, was not found to





exist freely in the pore-waters of SSK42/5 and 6; this is apparently attributable to the fact that its build-up to measurable quantities is generally debarred in natural environments due to high reactivity of this polythionate under the mediation of microbes (Kanao et al., 2007; Ghosh and Dam, 2009; Boden et al., 2010; Pyne et al., 2017, 2018) as well as naturally-occurring chemical substances such as sulfide (Schippers et al., 1999; Schippers and Jørgensen, 2001). From that

perspective, the *in situ* as well as *in vitro* geomicrobiological information unearthed in this study illustrates the power of meta-omics in discovering such invisible interfaces between the chemosphere and the biosphere that are almost impossible to decipher from preserved geochemical records alone.

**Supplementary material**

Supplemental material for this article may be found with the digital version of this manuscript.

**Data availability**

All nucleotide sequence data have been deposited in NCBI Sequence Read Archive (SRA) or

GenBank under the BioProject accession number PRJNA309469: (i) the whole metagenome shotgun sequence datasets have the Run accession numbers SRR3646127 through SRR3646132, SRR3646144, SRR3646145, SRR3646147, SRR3646148, SRR3646150 through SRR3646153, SRR3646155 through SRR3646158, SRR3646160 through SRR3646165, and (ii) the metatranscriptome sequence dataset has the Run accession number SRR7991972.


**Code availability.** All data analysis codes used in this study are in the published domain, and have been appropriately cited in the text.

**Author contributions**

W.G. conceived the study, designed the experiments, interpreted the results and wrote the paper. A.M led the entire SSK42 mission and all geochemical investigations therein. S.M. anchored the whole microbiological work, performed the experiments, analyzed and curated the data. S.B., T.M., M.J.R. and C.R. performed microbiological experiments and data analysis. S.F. and A.P. performed geochemical experiments. All authors read and vetted the manuscript.


**Acknowledgements**

Financial support for conducting the microbiological studies was provided by given by Bose Institute via internal faculty grants and Earth System Science Organization, Ministry of Earth Sciences (MoES), Government of India (GoI) via grant number MoES/36/00IS/Extra/19/2013. We

thank the Director CSIR-National Institute of Oceanography for facilitating the geochemical



studies and the research cruise SSK42 for acquisition of sediment cores. MoES (GAP2303) also funded the research cruise. All the support received from the CSIR-NIO Ship Cell members and the crew members of SSK42 is gratefully acknowledged. S.B. received fellowship from Bose Institute. SM got fellowship from Department of Science and Technology, GoI. MJR and C.R. got fellowship from University Grants Commission, GoI. S.F. received fellowships from Council of Scientific and Industrial Research, GoI.

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

**Legend of figures**

**Figure 1. Scatter plots showing the down-depth variations in the prevalence of key metabolic-types, or the pair-wise associations between the prevalence of the metabolic-types, along SSK42/5.** Parameters considered include (i) sediment-depth (in cmbsf), and percentages of metagenomic reads matching sequences from the genera of (ii) manganese-depositing, (iii) manganese-oxidizing, (iv) tetrathionate-reducing, (vi) tetrathionate-forming and (vii) tetrathionate-oxidizing bacteria. Plots corroborated by Pearson correlation coefficient (CC) and/or Spearman rank correlation coefficient (RCC) values ≥ + 0.8 with P < 0.05 are shown in blue. Whereas none of the plots were corroborated by negative CC or RCC values numerically ≥ 0.8 with P < 0.05, those corroborated by positive/negative CC and/or RCC values numerically ≤ 0.8 are shown in black, irrespective of whether P is < 0.05. All CC and RCC values pertaining to the above plots are given in Table S6.

**Figure 2. Scatter plots showing the down-depth variations in the prevalence of key metabolic-types, or the pair-wise associations between the prevalence of the metabolic-types, along SSK42/6.** Parameters considered include (i) sediment-depth (in cmbsf), and percentages of metagenomic reads matching sequences from the genera of (ii) manganese-depositing, (iii) manganese-oxidizing, (iv) tetrathionate-reducing, (vi) tetrathionate-forming and (vii) tetrathionate-oxidizing bacteria. Plots corroborated by Pearson correlation coefficient (CC) and/or Spearman rank correlation coefficient (RCC) values ≥ + 0.8 with P < 0.05 are shown in



blue. Whereas none of the plots were corroborated by negative CC or RCC values numerically ≥ 0.8 with $P < 0.05$, those corroborated by positive/negative CC and/or RCC values numerically ≤ 0.8 are shown in black, irrespective of whether P is $< 0.05$. All CC and RCC values pertaining to the above plots are given in Table S7.

**Figure 3. Metabolic transformations of thiosulfate and/or tetrathionate by representative strains of the various species-level entities isolated from 275 cmbsf of SSK42/6.** (**A** through **D**) thiosulfate to tetrathionate conversion by *Halomonas* sp 15WGF, *Methylophaga* sp. SBPC3*, Pseudomonas* sp. SBBB and *Stenotrophomonas* sp. SBPC3 respectively. (**E** and **F**) thiosulfate to tetrathionate conversion and tetrathionate oxidation to sulfate by *Pusillimonas* sp. SBSA

respectively. (**G**) tetrathionate oxidation to sulfate by *Halothiobacillus* sp. SB14A. (**H**) tetrathionate reduction to thiosulfate by *Enterobacter* sp. RVSM5a.

—●—, —▲— and —▼— and denote the concentration of sulfur (mM S) in the spent media, at any time-point of incubation in the form of thiosulfate, sulfate and tetrathionate respectively.

—■— denotes the pH of the spent medium at any given time point of incubation.


**Figure 4.** Schematic diagram showing the network of biotic and abiotic process that are potentially involved in the formation and transformation of tetrathionate in the Arabian Sea OMZ sediments.






**Table 1.** Concentrations of sulfate[1], sulfide[1], pyrite, iron[2], manganese[2] and thiosulfate in the individual sample-sites of SSK42/5 and SSK42/6.

**SSK42/5**

| Sediment-depth (cmbsf) | Sulfate (mM) | Sulfide (µM) | Pyrite (wt %) | Iron (ppm) | Manganese (ppm) | Thiosulfate (µM) |
|---|---|---|---|---|---|---|
| 1 | 28.8356 | 62.0819 | 0.05 | 9847 | 78.1 | 0.98 |
| 15 | 27.9356 | 57.264 | 0.06 | 13158 | 83.07 | 0.99 |
| 30 | 25.7704 | 59.3288 | 0.28 | - | - | - |
| 45 | 23.1922 | 61.3936 | 0.47 | - | - | 0.99 |
| 60 | 22.4571 | 99.2484 | 0.46 | - | - | 1.5 |
| 75 | 19.7155 | 194.2296 | 0.57 | - | - | - |
| 90 | 17.2669 | 313.3001 | 0.68 | - | - | 4.2 |
| 105 | 17.6498 | 427.0022 | 0.60 | - | - | - |
| 120 | 16.2797 | 226.5782 | 0.70 | 9574 | 74.3 | 4.0 |
| 135 | 15.8326 | 254.9349 | 0.57 | - | - | - |
| 145 | 14.7989 | 296.7817 | 0.58 | 9842 | 87.1 | 3.1 |
| 160 | 13.0228 | 270.0768 | 0.68 | - | - | 3.5 |
| 175 | 13.0084 | 183.9055 | 0.61 | - | - | - |
| 190 | 11.043 | 154.3099 | 0.75 | - | - | 3.2 |
| 205 | 10.5552 | 203.8653 | 0.83 | - | - | - |
| 220 | 9.6447 | 61.3936 | 0.74 | - | - | 1.1 |
| 235 | 8.3577 | 54.5109 | 0.49 | - | - | - |
| 250 | 6.6526 | 58.6405 | 0.50 | 17234 | 171.9 | 1.2 |
| 280 | 4.3842 | 54.5109 | 0.86 | - | - | - |
| 295 | 7.46 | 51.7578 | 0.80 | 12329 | 147.2 | 1.1 |

**SSK42/6**

| Sediment-depth (cmbsf) | Sulfate (mM) | Sulfide (µM) | Pyrite (wt %) | Iron (ppm) | Manganese (ppm) | Thiosulfate (µM) |
|---|---|---|---|---|---|---|
| 1 | 27.6718 | 0 | 0.11 | 9302 | 87.7 | 0 |
| 15 | 25.4969 | 0 | 0.06 | - | - | 0 |
| 30 | 26.6722 | 321.3249 | 0.14 | - | - | 2.5 |
| 45 | 23.1086 | 603.253 | 0.37 | - | - | 5.2 |
| 60 | 20.8097 | 586.3688 | 0.75 | - | - | 4.7 |
| 75 | 18.829 | 710.1869 | 0.90 | - | - | - |
| 90 | 17.2731 | 1070.385 | 0.64 | 9374 | 71.1 | 8.2 |
| 105 | 16.4295 | 507.5754 | 1.02 | - | - | - |
| 120 | 15.3848 | 772.0959 | 1.09 | 10722 | 77.1 | 7.5 |
| 135 | 14.4313 | 704.5588 | 0.58 | 12312 | 105 | 7 |
| 145 | 13.4006 | 1172.7375 | 0.66 | - | - | - |
| 160 | 12.1834 | 1037.6632 | 0.55 | - | - | 8.5 |
| 175 | 10.1722 | 1071.4318 | 0.62 | - | - | 8.1 |
| 190 | 9.296 | 947.6137 | 0.29 | - | - | - |
| 205 | 7.4672 | 1341.5804 | 0.39 | - | - | - |
| 220 | 5.6191 | 1274.0432 | 0.59 | - | - | 11.1 |
| 235 | 4.1801 | 1240.2747 | 0.32 | - | - | - |
| 250 | 2.0497 | 2010.2769 | 0.45 | - | - | - |
| 265 | 0.8708 | 970.1261 | 0.49 | 9232 | 105 | 6.8 |
| 280 | 0.2806 | 1217.7623 | 0.38 | 10359 | 116.9 | 5.5 |
| 295 | 0.4786 | 1116.4566 | 0.70 | - | - | - |

[1]    Data for pore-water sulfate and sulfide concentrations were taken from Fernandes et al. (2018); values for all the other chemical concentrations were determined in this study.

[2]    Concentrations given for iron and manganese were determined for bulk sediment-samples, which included both the solid-phases and pore-waters; all other concentrations given are for pore-waters extracted from individual sediment-samples.

40



**Table 2.** Bacteria isolated from 275 cmbsf of SSK42/6, and their tetrathionate-metabolizing properties.

| | Bacteria isolated in ASWT | | Bacteria isolated in ASWTY | Bacteria isolated in MSTY | | | Bacteria isolated in RVTr |
|---|---|---|---|---|---|---|---|
| | *Halothiobacillus* sp. | *Methylophaga* sp. | *Halomonas* sp. | *Pseudomonas* sp | *Stenotrophomonas* sp. | *Pusillimonas ginsengisoli* | *Enterobacter* sp. |
| Identification up to lowest taxonomic level possible | *Halothiobacillus* sp. | *Methylophaga* sp. | *Halomonas* sp. | *Pseudomonas* sp | *Stenotrophomonas* sp. | *Pusillimonas ginsengisoli* | *Enterobacter* sp. |
| Total number of strains isolated for the species-level cluster | 6 | 2 | 2 | 2 | 1 | 2 | 4 |
| Name of the representative strain | SB14A | SB9B = MTCC12599 | 15WGF = MCC3301 | SBBB = MTCC12600 | SBPC3 | SBSA = MTCC12558 | RVSM5a |
| 16S rRNA gene sequence accession number of the representative strain | LN999387 | LN999390 | LT607031 | LN999396 | LN999400 | LN999398 | MH593841 |
| Tetrathionate-metabolizing phenotype (medium in which phenotype was tested) | (i) Tetrathionate to sulfate (ASWTr) | (i) Thiosulfate to tetrathionate (ASWTM) | (i) Thiosulfate to tetrathionate (ASWTY) | (i) Thiosulfate to tetrathionate (MSTY) | (i) Thiosulfate to tetrathionate (MSTY) | (i) Thiosulfate to tetrathionate (MSTY) (ii) Tetrathionate to sulfate (MSTrY) | (i) Tetrathionate to thiosulfate (RVTr) |

45




**Figure 1**

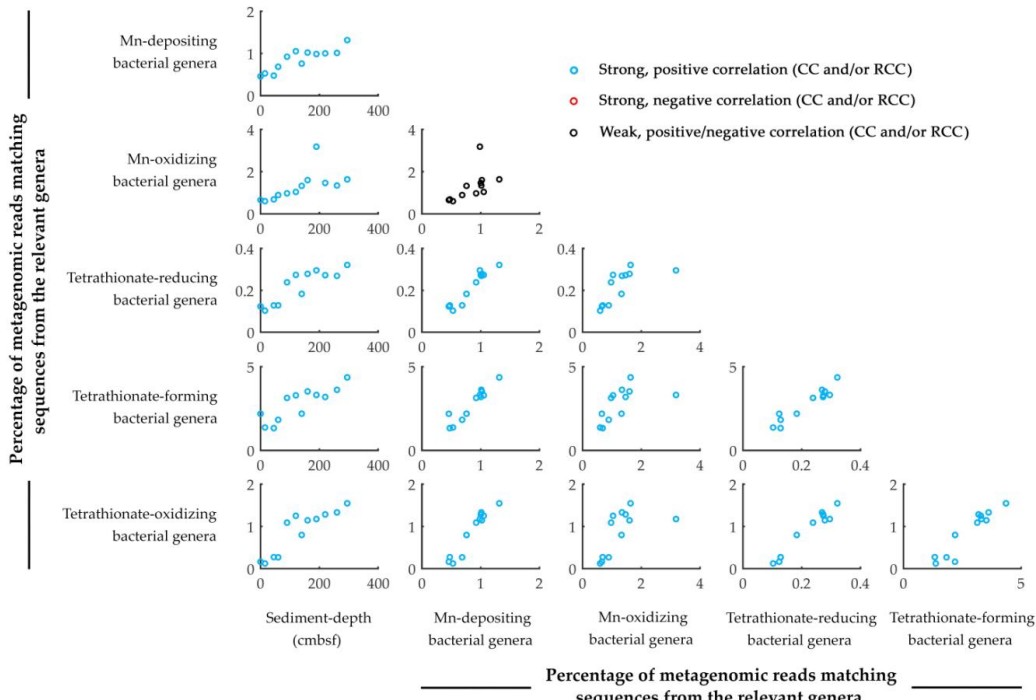

850

**Figure 2**

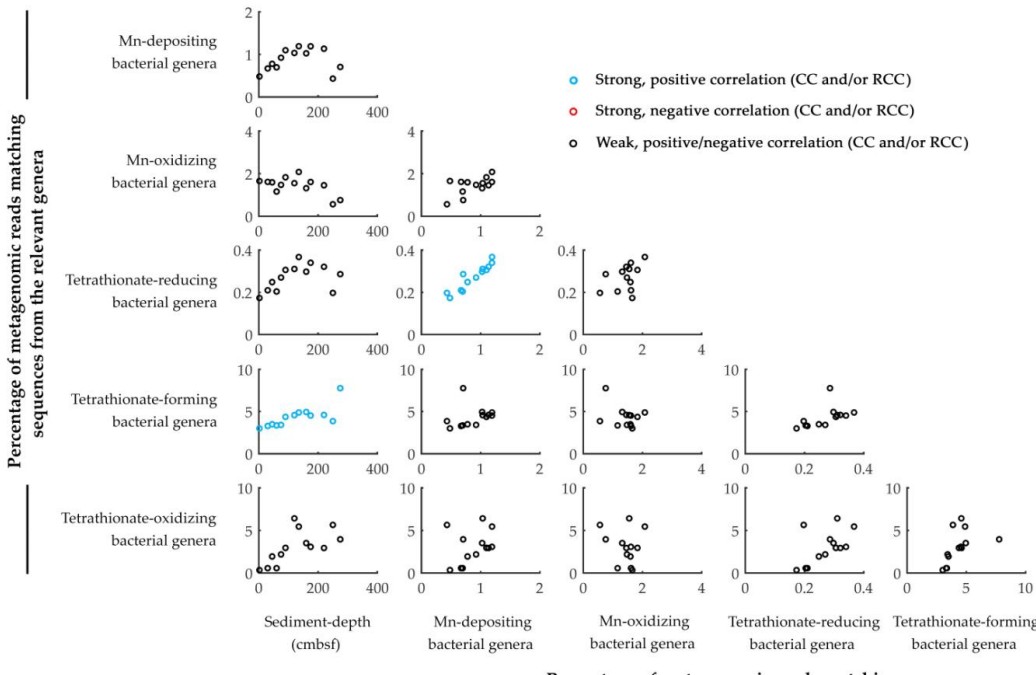





**Figure 3**

855

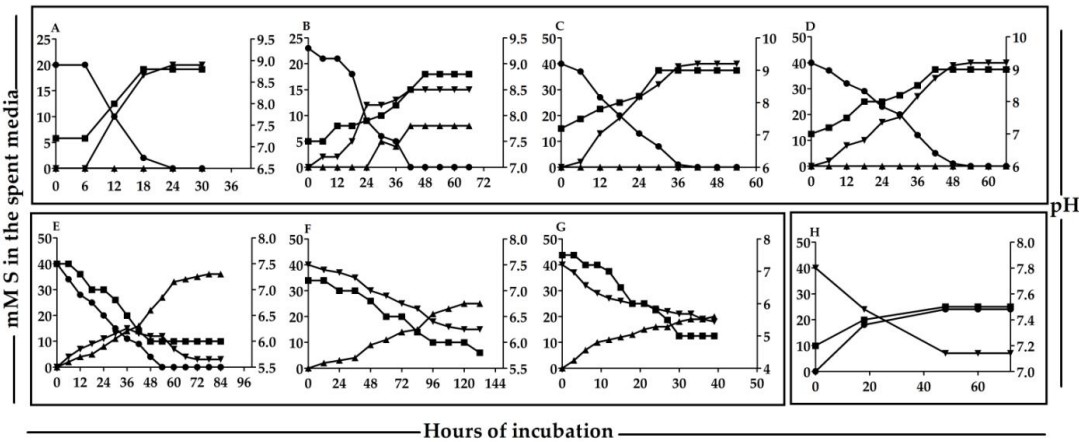

**Figure 4**

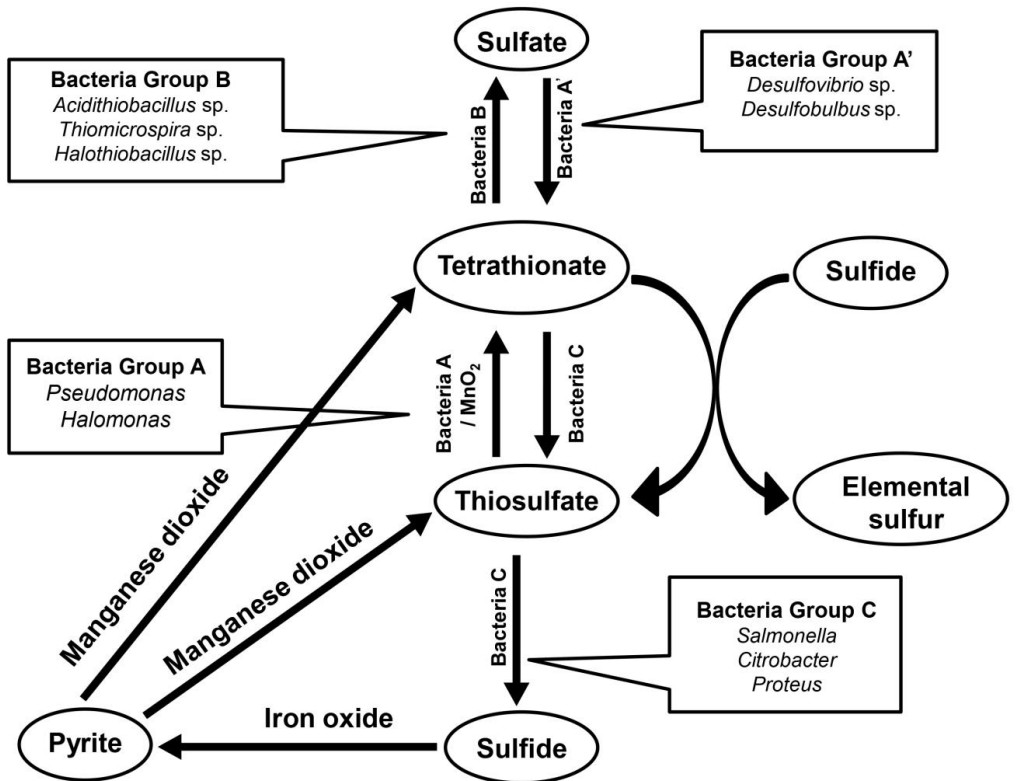

860