# Peer review of "Cryptic role of tetrathionate in the sulfur cycle: A study from Arabian Sea oxygen minimum zone sediments"

_Biogeosciences, 2019_

## Referee Comment (RC1) · Anonymous Referee #1 · 16 Jul 2019

Cryptic role of tetrathionate in the sulfur cycle: A study from the Arabian Sea oxygen minimum zone sediments.

General comments:

The paper investigated the role of tetrathionate as an intermediate in the redox cycling of sulfur in sediments from the oxygen minimum zone from the Arabian Sea. Using metagenomics approach, the authors find the presence of tetrathionate generating, oxidizing or reducing genes and identify bacteria potentially responsible for such processes. Through slurry incubations, the authors show the involvement of tetrathionat in the microbial sulfur cycle in these sediments. Tetrathionate itself was not detected

in-situ most, likely due to its reactivity. The authors propose pyrite and/or thiosulfate as potential sources of tetrationate, which is subsequently oxidized or reduced in the system.

The sampling approach is rather unusual (subsampling of oxidation critical subsamples from split-cores, see comments below). In addition, the description of the different subsamples are not entirely clear to me, which is most likely a formulation issue (see details below).

Description of the analytical methods are not precise (see detailed comments below).

Large parts of the text, especially in the results and discussion part should be rewritten and be more concise. The manuscript contains unnecessary text and phrases, which make reading complicated. Many sentences are too long and sometimes the grammar is not correct such that understanding is in parts not possible. Some examples (but not all) are pointed out/detailed below.

The figures should be better implemented and explained in the text where appropriate. Downhole analysis of chemical species could be visualized in a depth plot to provide a quicker overview for the reader. The data table can be part of the supplement. Results from the slurry incubations could be presented in an additional figure instead of (or in addition to) the tables. This would help understanding the complex results. From reading it seams very random when and in which samples e.g. tetrationate is oxidized and at what rates. I strongly suggest splitting results and discussion, this would help to sort out the text and help the reader understanding the story. There is also almost no discussion of the results rather than a presentation, e.g. there is no discussion of the determined rates and what they indicate etc... .

Collectively, I think this manuscript needs a major overhaul with focus on precise description of the sampling and methods and separation of results and methods including a proper and streamlined discussion, before the scientific merit can be judged. The extend of required rewriting including methods, results and discussion extends

what is justifiable as a revision. However, I would emphasis a re-submission as a new manuscript once rewritten.

In the following I provide many details, but this may not be complete.

Specific comments (incl. few technical comments):

Abstract

Line 25: introduce msbl, also: no dash between number and unit (msbl) here.

Lines 27/28: I suggest to be more precise and name the processes instead of generally speaking about "these metabolisms" or "these processes"

Line 29: Provide conditions of the incubations under which you could observe tetrathionate generation or turnover. (What types of slurry-incubations, i.e. with amendment of tetrathionate or thiosulfate... etc.)

Lines 31/32: Can you calculate a molar concentration or g/sed for iron and manganese instead of giving ppm

Line 34: instead of "converted" use "oxidized" here and throughout the manuscript (similarity use "reduced" if applicable)

Line 35: delete "back"

Line 35: avoid "0" as a concentration, it reads odd. 0 means absence, so 0-2 mM present is wrong as 0 means not present. You could write e.g. up to 2 mM

Introduction

Line 45: delete "running"

Line 48: delete "So"

Line 54: Delete: "In this context" – unnecessary

Line 55: replace/reformulate "seldom appreciated" by rarely investigated or similar

Material and Methods

Line 77: delete "the"

Line 78: delete "on which the present study is based" (unnecessary text)

Lines 81ff: The sampling strategy is unusual. Oxidation sensitive sample were collected after splitting the core into two halves. To prevent oxidation a shower of N2 was applied. How was this realized to ensure that no oxidation occurred? Usually smaller hole round core sections are subsamples inside an anaerobic camber (glove box) or subsamples are taken with cut-off syringes via small holes cut in the side of a liner or alternative from fresh cuts during sectioning. Al halve split exposes large areas to air even though somehow a N2 shower was installed this sees quite unusual. Was this split done at the entire 3 m core? How was a N2 shower over the 3-m length maintained during the sampling of the 10 – 20 subsamples from each core?

What are the "adequate measures" to avoid contamination? Does this refer only to the use of sterile spatulas? Were the sample bottles autoclaved?

Line 92: fractions means subsamples ?

Line 93: ", while one fraction, each for chemistry . . ." you mean two subsamples, one for chemistry and one for microbiology?

Line 99: ion chromatography

Line 101: what is the number in brackets? A catalog number? (should be removed).

Line 106: "passed through . . .membranes" – the samples were probably "filtered" – with syringe filters?

Line 108: Please more details on the method calibration: What standards were used for calibration, what calibration, how many points? External? What does "sample reproducibility," mean analytical precision? How was this determined, by how many replicate measurements of the same sample? The value should be given in molar

concentration if for a specific sample or in RSD (%) if it refers to the precision of the method itself.

Line 129 ff: for the determination of AVS and CRS fractions, original literature should be cited. How was $Ag_2S$ quantified, gravimetrically?

Line 261: 0% partial pressure? Pressure unit is not percent. Also 0 probably means anoxic?

Line 268ff: This is the standard cline protocol, which is widely used and generally accepted - not necessary to describe the principle.

Line 286: what does serially diluted mean?

Line 288: here and elsewhere, please use until instead of till. As this is a scientific article, and till is considered to be informal which should be avoided.

Line 289: "pure-plates" ?

Results and discussion

Line 327 "relevant (microorganisms)" ?

Line 329-334: Very long sentence- almost not understandable: consider splitting and rewriting.

Line 334 "were found to contain" – shorter "contained"

Line 341: Unpublished data should not be cited if not necessary. Here are 5 other references given. The reference to unpublished data is unnecessary.

Line 391-395: Long and unclear sentence.

Line 406 – 411: Long and unclear sentence.

Line 423: The discussion refers here to another unpublished paper. The suggestion/discussion here is based on unpublished data from the authors. Such data should

either be included in this manuscript or published first. Alternatively, the results should be discussed in the light of other already published studies. Otherwise, this discussion is not solid.

Lines 425 ff. The results of the incubation appear very unsystematic or random. A figure could help for an overview. The writing is also not precise, i.e. the "samples" do not convert thiosulfate to tetrathionate . . . conversion was observed in the samples or the organisms in the sample convert the species. . .

Line 430ff: "In contrast, . . ." The sentence is very long. Also, there is no "contrast" obvious. "free and detectable" is unnecessary.

Lines 434ff: the samples do not metabolize. Organisms have a metabolism but not a sediment samples.

In this entire section is not clear how the rates were determined. A figure might help.

---

## Referee Comment (RC2) · Anonymous Referee #2 · 21 Jul 2019

Review of BG-2019-248 July 21, 2019

The authors investigate the capacity of microorganisms in sediments from the Arabian Sea to metabolize redox-intermediate sulfur species. This is an important question with significance for understanding carbon turnover in sediments as well as for interpreting sedimentary records, in which these rapid processes are often invisible. The study relies primarily on the successful culturing of organisms with at least the facultative capacity to metabolize tetrathionate, and on metagenomic and metatranscriptomic datasets for a series of depths in two cores. The data are interesting and would fill an important gap in the literature, if the authors can address possible methodological

concerns below. A more serious issue, however, appears in the argument regarding tetrathionate production from pyrite. This idea is not present in any of the three papers cited, which evidences a serious misunderstanding about this process and causes me to recommend rejection of the paper in its current form. Additional comments and suggestions follow.

{Note – I am not able to speak to the appropriateness of the metagenomic methods described here, since this is outside my area of expertise.}

General comment – for reading clarity, there are many places were codes could be simplified. You only have two cores – they could be referenced as A and B, or by collection depth, much more readably than SSK42/5 and SSK42/6. The names of slurry media types are similarly difficult to read (e.g. was 'T' tetrathionate or thiosulfate?).

Bulk sediment porewater geochemistry – please consider converting information in Table 1 to a pair of depth profile figures, one for each site. Please provide an appropriate number of significant figures for your data (you are currently reporting sub-nM precision).

Thiosulfate and sulfide concentrations – No details are provided about how porewater sampling was conducted, or how porewater samples were chemically preserved. (Were zinc acetate, bromobimane, or other preservatives used? How much time elapsed between collection and IC analysis? What transformations might have occurred among your redox-sensitive dissolved species?)

Depth trends - Figure 1 seems to show that all five categories of relevant genera increase in their relative proportion smoothly with depth, which seems like it could be to first order a change in dilution of the microbial population with organisms that prefer the shallower sediments (aerobes or those that subsist on fresh, relatively shallow organic matter). The significance of this first-order depth trend needs to be discussed separately in the discussion. Standard depth profiles of the five genera groups would be much easier for the reader to interpret.

A more thorough description of how cores 42/5 and 42/6 differ sedimentologically would improve the analysis, especially since the key correlations from Figs 1 and 2 are only strongly significant in one of the cases.

Line 267 – I read this statement to mean that some representatives of each of these groups have been observed to cycle tetrathionate. Clearly, though, that does not apply to all members of these broad taxonomic groups, which makes it very difficult to tell whether the organisms might generally or facultatively cycle tetrathionate or something else entirely. Are statistics available on what fraction of, say, Salmonella has the genetic machinery for tetrathionate conversion?

Line 423 –Unless the data is included here, your own unpublished conclusions can't be cited like this.

Much of section 3.3 is a reporting of results without context or discussion, which is challenging to parse for key points – consider separating out your results. Figures would be very helpful. How do these rates compare with other culture studies or with the size of the porewater pools? What are these depth patterns? What do you want your reader to gain from this information?

The title of section 3.3 isn't quite true – these experiments show the presence of living organisms that can at least facultatively do these metabolisms; it does not show that they are actively conducting these metabolisms in-situ. Can you integrate this conclusion with your RNA results or show actual in-situ abundances of your cultured organisms?

Line 385 – intracellular vs extracellular tetrathionate. Most of these examples of tetrathionate producers generate tetrathionate as an intracellular intermediate species during metabolism. Please discuss how you envision other members of the microbial community accessing these species.

Line 445-453 - How do you possibly get as many as six separate samples that have

identical observed rates (e.g., of 141ïĆś 1 ïA■mol S/d/g)?

Line 492 – The authors cite three papers to support a link between MnO2 cycling and pyrite oxidation to tetrathionate and other dissolved species. I have no idea what the authors are referencing, which is troubling and forces me to recommend rejection. The Berner and Petsch paper from 1998 does not include the words manganese, thiosulfate, or tetrathionate. There is similarly no mention of MnO2 in the Luther 1991 paper. And, although the Jørgensen and Bak paper discusses manganese, it is in the context of "manganese or iron oxides" which could similarly be used as electron acceptors, not anything about pyrite oxidation.

Line 498 – The entire argument for pyrite oxidation by MnO2 appears to be that there is detectable Mn in the sediments. (Basically all sediments have this??) There must be more one could say on this topic. . . depth trends? Differences between the two cores? Comparison with typical sediment Mn concentrations? Otherwise I'd leave the Mn discussion out.

I would certainly not use this discussion to conclude that "Pyrites (via abiotic reaction with MnO2) and thiosulfate (via chemolithotrophic oxidation by members of the bacterial group designated as A in Fig. 4) are apparently the main sources of tetrathionate". This has not been demonstrated.
* * *

---

## Referee Comment (RC3) · Anonymous Referee #2 · 23 Jul 2019

original comment: "Line 492 – The authors cite three papers to support a link between MnO2 cycling and pyrite oxidation to tetrathionate and other dissolved species. I have no idea what the authors are referencing, which is troubling and forces me to recommend rejection. The Berner and Petsch paper from 1998 does not include the words manganese, thiosulfate, or tetrathionate. There is similarly no mention of MnO2 in the Luther 1991 paper. And, although the Jørgensen and Bak paper discusses manganese, it is in the context of "manganese or iron oxides" which could similarly be used as electron acceptors, not anything about pyrite oxidation.

SM: WE ARE EXTREMELY SORRY for this "copy-paste" goof-up committed in the

haste of submitting multiple manuscripts within the same time-window! The actual reference that should have been used is Schippers, A., and Jørgensen, B. B.: Oxidation of pyrite and iron sulfide by manganese dioxide in marine sediments, Geochim. Cosmochim. Acta., 65, 915-922, https://doi.org/10.1016/S0016-7037(00)00589-5, 2001. (Please note that this was already included in the Reference list and cited in another context of Discussion). This paper should have also been cited in this particular context of tetrathionate production from pyrite, instead of the three irrelevant references that mistakenly got inserted. Please see Line numbers 4-8 of the Abstract itself of Schippers and Jørgensen, 2001, which clearly states that "FeS2 and iron sulfide (FeS) were oxidized chemically at pH 8 by MnO2 but not by nitrate or amorphic Fe(III) oxide. Elemental sulfur and sulfate were the only products of FeS oxidation, whereas FeS2 was oxidized to a variety of sulfur compounds, mainly sulfate plus intermediates such as thiosulfate, trithionate, tetrathionate, and pentathionate. Thiosulfate was oxidized by MnO2 to tetrathionate while other intermediates were oxidized to sulfate."

Current Response: Thank you for clarifying the correct references here, your thinking is far clearer now. Although this reference does describe pyrite oxidation via MnO2, I am still unconvinced that it is relevant to the sediments in the current study. If one reads beyond the abstract of that paper, one also finds that "Below 7.5 cm, where the content of Mn did not exceed 0.2% (w/w), a dissolution of FeS2 was not detectable." One also finds that the abiotic incubations produced intermediately only for days to weeks and not longer, decreasing rapidly with depth. Manganese concentrations in the current paper (71-172 ppm) are orders of magnitude lower than the threshold for activity reported before, and there are no depth trends in either pyrite or MnO2 discussed, or porewater metal ion data, that might support this mechanism as active. Purported Mn-driven oxidation also appears to increase, rather than decrease, with depth, in contrast with the prior report. I do not think pyrite is a source of dissolved S species in this system; stronger evidence is required.

Allied key comment: I would certainly not use this discussion to conclude that "Pyrites

(via abiotic reaction with MnO2) and thiosulfate (via chemolithotrophic oxidation by members of the bacterial group designated as A in Fig. 4) are apparently the main sources of tetrathionate". This has not been demonstrated.

Our response: Chemolithotrophic conversion of thiosulfate to tetrathionate by members of the bacterial genera Pseudomonas and Halomonas (designated as A in Fig. 4) has been experimentally demonstrated - we have shown such isolates of both Pseudomonas and Halomonas which are capable of chemolithotrophically converting thiosulfate to tetrathionate in vitro. Furthermore, when metagenomic sequence data obtained for each of the 25 distinct sediment-samples of SSK42/5 and 6 were assembled and annotated individually, 23 out of the 25 contig-collections obtained were found to contain genes for tetrathionate formation (namely, genes encoding subunits of the thiosulfate dehydrogenases TsdA that converts thiosulfate to tetrathionate; see Denkmann et al., 2012; Pyne et al., 2018) [Table S3]. Whole metatranscriptome sequencing and analysis for the 275 cmbsf sediment-sample of SSK42/6 also revealed the gene-catalog obtained via annotation of the assembled contigs to encompass homologs of the thiosulfate dehydrogenase gene tsdA [Table S19]. These data clearly supported the potential in situ functionality (metabolically) of thiosulfate to tetrathionate converting bacteria.

Current Response: I do not dispute that thiosulfate-to-tetrathionate conversion was demonstrated and is quite intriguing; the piece of your claim that has not been demonstrated is related to pyrite. Without showing any depth trends, porewater metal ion data, or pyrite-specific (tracer) incubations, there is no data evidencing the involvement of pyrite.

---

## Author Comment (AC1) · 23 Jul 2019

Dear Reviewer 2 and Dear Dr. Treude, We just felt the urgency of writing this inter-mittent mail so as to rectify and clarify the most serious issue that appeared in the argument regarding tetrathionate production from pyrite, and that which caused a seri-ous misunderstanding about this process forcing Reviewer 2 to recommend rejection of the paper in its current form. While we keep working on an elaborate revision following all the other suggestions and criticisms offered by Reviewer 2 and others, below please find our responses to the key concern in hand, and kindly consider this amendment in your decision-making exercise. Yours, with regards Wriddhiman

Main comment: The authors investigate the capacity of microorganisms in sediments from the Arabian Sea to metabolize redox-intermediate sulfur species. This is an important question with significance for understanding carbon turnover in sediments as well as for interpreting sedimentary records, in which these rapid processes are often invisible. The study relies primarily on the successful culturing of organisms with at least the facultative capacity to metabolize tetrathionate, and on metagenomic and metatranscriptomic datasets for a series of depths in two cores. The data are interesting and would fill an important gap in the literature, if the authors can address possible methodological concerns below. A more serious issue, however, appears in the argument regarding tetrathionate production from pyrite. This idea is not present in any of the three papers cited, which evidences a serious misunderstanding about this process and causes me to recommend rejection of the paper in its current form. Additional comments and suggestions follow. Line 492 – The authors cite three papers to support a link between MnO2 cycling and pyrite oxidation to tetrathionate and other dissolved species. I have no idea what the authors are referencing, which is troubling and forces me to recommend rejection. The Berner and Petsch paper from 1998 does not include the words manganese, thiosulfate, or tetrathionate. There is similarly no mention of MnO2 in the Luther 1991 paper. And, although the Jørgensen and Bak paper discusses manganese, it is in the context of "manganese or iron oxides" which could similarly be used as electron acceptors, not anything about pyrite oxidation.

Our response:

WE ARE EXTREMELY SORRY for this "copy-paste" goof-up committed in the haste of submitting multiple manuscripts within the same time-window! The actual reference that should have been used is

Schippers, A., and Jørgensen, B. B.: Oxidation of pyrite and iron sulfide by manganese dioxide in marine sediments, Geochim. Cosmochim. Acta., 65, 915-922, https://doi.org/10.1016/S0016-7037(00)00589-5, 2001. (Please note that this was already included in the Reference list and cited in another context of Discussion). This

paper should have also been cited in this particular context of tetrathionate production from pyrite, instead of the three irrelevant references that mistakenly got inserted. Please see Line numbers 4-8 of the Abstract itself of Schippers and Jørgensen, 2001, which clearly states that "FeS2 and iron sulfide (FeS) were oxidized chemically at pH 8 by MnO2 but not by nitrate or amorphous Fe(III) oxide. Elemental sulfur and sulfate were the only products of FeS oxidation, whereas FeS2 was oxidized to a variety of sulfur compounds, mainly sulfate plus intermediates such as thiosulfate, trithionate, tetrathionate, and pentathionate. Thiosulfate was oxidized by MnO2 to tetrathionate while other intermediates were oxidized to sulfate."

Allied key comment:

I would certainly not use this discussion to conclude that "Pyrites (via abiotic reaction with MnO2) and thiosulfate (via chemolithotrophic oxidation by members of the bacterial group designated as A in Fig. 4) are apparently the main sources of tetrathionate". This has not been demonstrated.

Our response:

Chemolithotrophic conversion of thiosulfate to tetrathionate by members of the bacterial genera Pseudomonas and Halomonas (designated as A in Fig. 4) has been experimentally demonstrated - we have shown such isolates of both Pseudomonas and Halomonas which are capable of chemolithotrophically converting thiosulfate to tetrathionate in vitro. Furthermore, when metagenomic sequence data obtained for each of the 25 distinct sediment-samples of SSK42/5 and 6 were assembled and annotated individually, 23 out of the 25 contig-collections obtained were found to contain genes for tetrathionate formation (namely, genes encoding subunits of the thiosulfate dehydrogenases TsdA that converts thiosulfate to tetrathionate; see Denkmann et al., 2012; Pyne et al., 2018) [Table S3]. Whole metatranscriptome sequencing and analysis for the 275 cmbsf sediment-sample of SSK42/6 also revealed the gene-catalog obtained via annotation of the assembled contigs to encompass homologs of the thiosulfate dehydrogenase gene tsdA [Table S19]. These data clearly supported the potential in situ functionality (metabolically) of thiosulfate to tetrathionate converting bacteria.

---

## Short Comment (SC1) · 24 Jul 2019

As an one of the corresponding author, I agree with the rev#2's observation that the $MnO_2$-$FeS_2$ interaction as a source of tertrathionate has been overstretched and indeed needs more experimentation to substantiate. We agree to curtail this particular discussion component and limit only to the observation and conclusion drawn from the micobialal studies.

---

## Referee Comment (RC4) · Anonymous Referee #3 · 2 Aug 2019

The manuscript describes the analysis of inorganic sulfur compound cycling microbial populations in two sediment cores from the Indian Ocean. Particular focus is on populations driving the metabolism of thiosulfate and tetrathionate (thiosulfate reducing/tetrathionate forming, tetrathionate reducing, and tetrathionate oxidising groups). The study used a range of geochemical measurements, slurry incubations, microbiology (isolation of sulfur cycling microorganisms and assessment of their capabilities to transform thiosulfate and tetrathionate) as well as molecular biological approaches (metagenomics and metatranscriptomics).

The biogeochemistry of sulfur compounds in sediments is a complex web of chemical

and biological transformations, there is a need to better understand the role of individual species of inorganic sulfur compounds as well as the metabolic pathways and microbial groups involved in their transformations. As such, this is a topic of high interest, especially if, as the title implies, some of the transformations may be of a cryptic (not easily identified) nature.

My overarching impression of the manuscript is that it is not easy to follow the story and that it would benefit from revising the structure. It is lacking a clear approach to the analysis and presentation of the data. Even starting in the introduction, I would suggest that, given the focus on the various enzymes being instrumental in the transformations of thiosulfate and tetrathionate, the introduction should provide a brief overview of the most important enzymes involved (and their encoding genes) and perhaps contain a schematic conceptual overview illustrating the most important points.

It would be beneficial and aid readability, if a clear overview of the basic findings was shown perhaps as depth profiles showing key chemical parameters of the cores under investigation.

With a view of the diversity and metagenomics analysis, I have two key criticisms: (i) revolving around the specific use of metagenomics read data for taxonomic assignment and (ii) extrapolating from that assignment to physiological properties of entire genera of bacteria. In that context, I have to say that I think it is a pity the authors did not carry out a diversity analysis of the sediment samples based on pooled 16S rRNA amplicon sequencing in parallel to the metagenomics/metatranscriptomics, because the ribosomal RNA gene survey would provide a much better and more robust diversity analysis than the assignment of taxonomy based on random metagenomic reads. Although a taxonomic assignment of a random metagenomic read is possible, it is fraught with major uncertainty, unless a closely related organism's genome is available in a database. As that is not the case for the vast majority of microorganisms found in nature at present, the taxonomic assignment of metagenomic reads is a bound to provide unreliable/unresolved taxonomies and lead to poor estimates of the abundance of

specific types of bacteria. This affects data shown in Tables S8-13 as well as Fig 1 and 2.

My second criticism is the assumption in the paper that entire genera of bacteria always share specific physiological capacities with respect to the sulfur transformations of interest. While this is true for some genera (and a parameter used in systematics), for many genera this is not necessarily the case. Therefore, suggesting that genus A or B are tetrathionate producing bacteria or -oxidising bacteria, will likely overestimate the abundance of that specific metabolic type based on that assumption.

On the other hand, the metagenomics data can reveal the abundance of specific types of genes, as has been done here, and potentially identify the types of bacteria potentially contributing to the cycling of specific compounds. Too much of the discussion of inorganic sulfur metabolism in these sediments is based on the broad assumption of taxonomy, and too little is made of the specific genes found and listed in various supplementary tables. There are still limitations of our understanding of the genetics of sulfur transformations and it would be useful to perhaps illustrate whether the enzymes/genes driving specific transformations of sulfur compounds in some of the taxa mentioned (eg Salmonella) have actually been identified. Very little is done with the metatranscriptome data, it only gets a few mentions, but there is no clear overview of what has been found in which layer, how many reads were analysed and generally which bacteria were transcriptionally active with respect to sulfur cycling.

Regarding the transformations measured in slurry experiments, there needs to be a more complete reporting of the activities measured (or not) in all sediment samples. This should be shown comprehensively, not as currently done in Tables S14-S21, which suggest that only a few subsamples had certain activities. I am also not convinced that a 30-day incubation period of the slurries, some incubating sediments from a completely anoxic system under aerobic conditions (!), is providing the sort of activity data that would be supportive of suggesting that these key biological reactions are linked up in a cryptic cycle. The mentioning of alternative sources of oxygen from cryptic sources

such as perchlorate is pointing to an unpublished study by the same authors. If this is crucial for the understanding of the functioning of this system and the microbiological activities required for the cryptic cycling of these compounds, these aspects should be incorporated here or the other study needs to be published. Alternatively, it is possible that the activities are due to facultative anaerobic bacteria in these sediments that have reverted back to an aerobic lifestyle given suitable incubation conditions and a 30-day period to wake up.

Specific comments

Introduction: The introduction would benefit from a description of relevant metabolic pathways and enzymes targeted by the analysis of this paper

Line 73: define mbsl

Line 75 what was the diameter of these cores?

Line 82: for ease of reading I suggest to always refer to both cores with the full abbreviation, not SSK42/5 and 6 but SSK42/5 and SSK42/6

Line 82: I find the description of the N2 shower lacking in detail and find it hard to understand how it would keep the exposed core adequately protected from oxygen.

Line 88: scraped

Line 145 define cmbsf , how thick was that sediment layer?

Line 147 and 190: pooling up, no 'up'

Line 213: kmer lenghts of . . .

Line 227: text string search?

Line 236/7: not every read would represent one of these functional groups. Please reword/revise or explain more clearly what you did. There are no 'genera' of this that and the other, they are genera that contain species some of which have certain physiological characteristics, but not necessarily all of them

Line 246 following: No context for why one would assess the aerobic metabolism of these compounds with samples from 275cm below the sediment surface where there is no oxygen.

Line 309: actually described below

Line 327: reword, genes do not have these activities, they encode enzymes that transform the compounds

Line 350: I do not think that all of marine Pseudomonas and Halomonas do that

Table S1 Please clarify what is meant with 1st or 2nd sample fraction

Table S4 Not a single DoxA encoding gene was identified in any of the samples. Should it perhaps not be listed in Table S4 accordingly? Please state what the significance of the yellow highlighting is in this and the other excel based supplementary tables

Table S6 Define what is meant with prevalence and how it is quantified Unclear what is meant with correlation of metabolic type with sediment depth, when depth is not quantified here

Tables S8 to S11 should have totals for the abundance of all types per depth

---

## Author Comment (AC2) · 20 Sep 2019

General comments:

Referee's Comment: The paper investigated the role of tetrathionate as an intermediate in the redox cycling of sulfur in sediments from the oxygen minimum zone from the Arabian Sea. Using metagenomics approach, the authors find the presence of tetrathionate generating, oxidizing or reducing genes and identify bacteria potentially responsible for such processes. Through slurry incubations, the authors show the involvement of tetrathionate in the microbial sulfur cycle in these sediments. Tetrathionate itself was not detected in-situ most, likely due to its reactivity. The authors propose pyrite and/or thiosulfate as potential sources of tetrathionate, which is subsequently oxidized or reduced in the system.

Authors' Response: We thank the Reviewer for his appreciation of the phenomenon unearthed in this study.

Referee's Comment: The sampling approach is rather unusual (subsampling of oxidation critical subsamples from split-cores, see comments below). In addition, the description of the different subsamples are not entirely clear to me, which is most likely a formulation issue (see details below). Description of the analytical methods are not precise (see detailed comments below).

Authors' Response: We agree that there were some deficiencies in our explanation of the sampling and analytical procedures, so in the revised manuscript we have now described these in a more detailed and scientifically explicit manner. Improvements in these regards can be identified through changes in the text shown in the Track Changes file of the manuscript and our responses to the similar points mentioned below in your line-by-line specific comments.

Referee's Comment: Large parts of the text, especially in the results and discussion part should be rewritten and be more concise. The manuscript contains unnecessary text and phrases, which make reading complicated. Many sentences are too long and sometimes the grammar is not correct such that understanding is in parts not possible. Some examples (but not all) are pointed out/detailed below.

Authors' Response: We have now addressed these concerns by removing the extraneous details, making the language lucid and sentences simple. We believe all these have added to the quality of the writing and comprehensibility of the underlying science.

Referee's Comment: The figures should be better implemented and explained in the

text where appropriate. Downhole analysis of chemical species could be visualized in a depth plot to provide a quicker overview for the reader. The data table can be part of the supplement. Results from the slurry incubations could be presented in an additional figure instead of (or in addition to) the tables. This would help understanding the complex results. From reading it seems very random when and in which samples e.g. tetrathionate is oxidized and at what rates. I strongly suggest splitting results and discussion, this would help to sort out the text and help the reader understanding the story. There is also almost no discussion of the results rather than a presentation, e.g. there is no discussion of the determined rates and what they indicate etc.

Authors' Response: We agree with these suggestions, and have now taken the following measures to ensure that the problems associated with the text and display items are all ameliorated. - More citations and explanations for the figures have been included in the text. - Analyses of chemical species along the sediment-cores have now been presented in the form of depth plots; accordingly, Table 1 is now moved from the main text to the Supplementary Information. - A new Figure (numbered as 4 in the revised manuscript) has now been incorporated for the results of the slurry incubation experiments. We believe that the visual impact of the data is now instrumental in an easier comprehension of the complex results pertaining to the formation and/or oxidation of tetrathionate, plus the reduction of tetrathionate. - Results and Discussion sections have now been split. Implications of all the current findings (results/data) have been explained adequately in the new Discussion section.

Referee's Comment: Collectively, I think this manuscript needs a major overhaul with focus on precise description of the sampling and methods and separation of results and methods including a proper and streamlined discussion, before the scientific merit can be judged. The extend of required rewriting including methods, results and discussion extends what is justifiable as a revision. However, I would emphas a re-submission as a new manuscript once rewritten.

Authors' Response: We have now overhauled the manuscript exactly as suggested.

Besides more streamlined description of the methods and sampling procedure, separation of Results and Discussion sections, and result-specific discussions, we have now included new lines of metagenomic, genomic and metatranscriptomic data (and their corresponding discussions) in the revised manuscript. New display items have also been provided for the existing data. Furthermore, we have now deleted a number of such geochemical data that were inexplicable under the current state of knowledge on the biogeochemical system explored in this study.

Referee's Comment: In the following I provide many details, but this may not be complete. Specific comments (incl. few technical comments):

Abstract Line 25: introduce msbl, also: no dash between number and unit (msbl) here.

Authors' Response: We agree, and have now fixed this problem.

Referee's Comment: Lines 27/28: I suggest to be more precise and name the processes instead of generally speaking about "these metabolisms" or "these processes"

Authors' Response: We agree that the use of precise names for anything technical is always preferable in scientific literature. However, since in the present case the words "microbial formation, oxidation and reduction of tetrathionate" are appearing twice within the same sentence we prefer to mention the proper nouns initially and use the pronoun "these metabolism" in the second occasion.

Referee's Comment: Line 29: Provide conditions of the incubations under which you could observe tetrathionate generation or turnover. (What types of slurry-incubations, i.e. with amendment of tetrathionate or thiosulfate: : : etc.)

Authors' Response: Here in the abstract, we have now mentioned the media compositions used in the slurry incubation experiments; more details are already there in the main text.

Referee's Comment: Lines 31/32: Can you calculate a molar concentration or g/sed for iron and manganese instead of giving ppm

Authors' Response: We agree that in any scientific literature molar concentrations are more preferable than ppm values. Incidentally, however, all data and discussions pertaining to the in situ concentrations of iron and manganese have now been removed from the revised manuscript following the suggestions of Reviewer 2.

Referee's Comment: Line 34: instead of "converted" use "oxidized" here and throughout the manuscript (similarity use "reduced" if applicable)

Authors' Response: We agree, now fixed.

Referee's Comment: Line 35: delete "back"

Authors' Response: Please note that the very preceding sentence reads as "Thiosulfate oxidation by chemolithotrophic bacteria prevalent in situ is the apparent source of tetrathionate in this ecosystem"; so, in this succeeding sentence, we think that it would be more appropriate to write "tetrathionate can be reduced back to thiosulfate" than simply say "tetrathionate can be reduced to thiosulfate".

Referee's Comment: Line 35: avoid "0" as a concentration, it reads odd. 0 means absence, so 0-2 mM present is wrong as 0 means not present. You could write e.g. up to 2 mM

Authors' Response: We agree, and have now written "up to" wherever it was applicable.

Introduction

Referee's Comment: Line 45: delete "running"

Authors' Response: We have now done as suggested.

Referee's Comment: Line 48: delete "So"

Authors' Response: Done.

Referee's Comment: Line 54: Delete: "In this context" – unnecessary

Authors' Response: We agree, now deleted.

Referee's Comment: Line 55: replace/reformulate "seldom appreciated" by rarely investigated or similar

Authors' Response: We agree, and have now reworded this as suggested.

Material and Methods

Referee's Comment: Line 77: delete "the"

Authors' Response: Here we are referring to two specific cores out of the total 8 collected on board SSK42. Moreover, the first mention of two SSK42 cores constituting the raw material of this study had already happened only seven lines ahead of this line, so the article "the" should be used here ahead of the words "two gravity cores".

Referee's Comment: Line 78: delete "on which the present study is based" (unnecessary text)

Authors' Response: We have now restructured this sentence in such a way as to introduce the readers to the core nomenclature and at the same time convey that out of the ten SSK42 cores, which are being studied and reported (from distinct biogeochemical perspectives) in a series of recent publications by our group, the 5th and 6th (already named in Fernandes et al., 2018 as SSK42/5 and SSK42/6) are the ones on which the present study is based.

Referee's Comment: Lines 81ff: The sampling strategy is unusual. Oxidation sensitive sample were collected after splitting the core into two halves. To prevent oxidation a shower of N2 was applied. How was this realized to ensure that no oxidation occurred? Usually smaller hole round core sections are subsamples inside an anaerobic camber (glove box) or subsamples are taken with cut-off syringes via small holes cut in the side of a liner or alternative from fresh cuts during sectioning. Al halve split exposes large areas to air even though somehow a N2 shower was installed this sees quite unusual. Was this split done at the entire 3 m core? How was a N2 shower over the 3-m length maintained during the sampling of the 10 – 20 subsamples from each core?

Authors' Response: We have now overhauled the portion dealing with sampling details in such a way that all confusions emanating from the previous text are now resolved. This said, it is noteworthy that answers to these questions were already there in our previous publication Fernandes et al., 2018, which also dealt with these SSK42 cores (albeit form other perspectives) and was cited amply throughout the manuscript, including the sampling-related section. We had therefore thought that further repetition of the details would be unnecessary and also potent causes of unintended self-plagiarism. However, now we understand that as an independent paper this manuscript should carry its own sampling details and have therefore brought back many of those details taking sufficient care of literary repetitiveness. For the records please note the following.

- In order to protect the ASOMZ sediment-samples from aerial oxidation, the entire cores were not split open into two D-shaped halves directly; instead only one ∼15-cm-long C-shaped part of PVC core-liner was removed at a time, as shown in a new Supplementary Figure added to the revised manuscript. We have now restructured the old sentences and written additional new lines to explain this in an unambiguous way. - The 15 cm length exposed at a time for sampling was constantly and closely showered with high-purity $N_2$ emitted from multiple nozzles fitted to multiple nitrogen-generators. This contrivance was sufficient to prevent atmospheric oxidation of the 15 cm exposed surface of the core. - Immediately after the C-shaped longitudinal part of the PVC core-liner was cut open, top one cm of the exposed surface was scrapped off along the core-circumference using sterile scalpels to eliminate potential contaminations from the core-liners' inner-surfaces and/or sea-waters through which the cores had passed. - Subsequently, to sample a particular sediment-depth of the core for microbiological studies, an approximately 5-mm-thick sediment-slice (spanning equally on either side of the core-height marking) was scooped out with a sterile scalpel and put into a sterile polypropylene bottle. The head-space of every sample-containing bottle was flushed with high-pure $N_2$, following which it was sealed with Parafilm - At the same time, for on board extraction of pore-waters, sediment-samples from a particular depth were

taken out by inserting sterile 50 ml cut off syringes deep inside the core cross-section, multiple times along the circumference on the exposed 'C half'; the samples were immediately collected in sterile 50 ml centrifuge tubes; and all these were carried out under focused streams of high-pure N2.

Referee's Comment: What are the "adequate measures" to avoid contamination? Does this refer only to the use of sterile spatulas? Were the sample bottles autoclaved?

Authors' Response: Immediately after the C-shaped longitudinal part of the PVC core-liner was cut open, top one cm of the exposed surface was scrapped off along the core-circumference using sterile scalpels to eliminate potential contaminations from the core-liners' inner-surfaces and/or sea-waters through which the cores had passed. All the samples vials, spatulas and bottles used for sampling were sterile. They were either autoclaved on board or purchased as sealed packs of gamma irradiation sterilized laboratory wares.

Referee's Comment: Line 92: fractions means subsamples?

Authors' Response: Yes, here fractions means subsamples.

Referee's Comment: Line 93: "while one fraction, each for chemistry : : :" you mean two subsamples, one for chemistry and one for microbiology?

Authors' Response: The Reviewer's understanding of the sentence is exactly same as what was meant.

Referee's Comment: Line 99: ion chromatography

Author Response- The type error is now fixed.

Referee's Comment: Line 101: what is the number in brackets? A catalog number? (should be removed).

Authors' Response: The column model number has now been deleted. However, we think the IC detector model number would be important.

Referee's Comment: Line 106: "passed through : : :membranes" – the samples were probably "filtered" – with syringe filters?

Authors' Response: Yes, volumes of the samples were already mentioned to be in the order of $\mu$L, so it could be filtered only using a syringe. Following your suggestion we have now added the word "filters" after the word "membrane", so this sentence in the revised manuscript now reads as "pore-water samples were ... passed through 0.22 $\mu$m hydrophilic polyvinylidene fluoride membrane filters".

Referee's Comment: Line 108: Please more details on the method calibration: What standards were used for calibration, what calibration, how many points? External? What does "sample reproducibility," mean analytical precision? How was this determined, by how many replicate measurements of the same sample? The value should be given in molar concentration if for a specific sample or in RSD (%) if it refers to the precision of the method itself.

Authors' Response: We agree, and have now provided more details for the method of IC calibration. Analytical grade thiosulfate IC Standard [ICS024, Sigma Aldrich, (St. Louis, MO, USA)] was used to prepare the calibration curve for quantification of this anion. Three different concentrations of thiosulfate, 0.5 $\mu$M, 5 $\mu$M and 20 $\mu$M, were measured for the construction of calibration curve by plotting peak height against concentration. Based on triplicate analyses of the standards, deviations from actual concentrations were found to be less than 2.5%.

Referee's Comment: Line 129 ff: for the determination of AVS and CRS fractions, original literature should be cited. How was Ag2S quantified, gravimetrically?

Authors' Response: Ag2S was quantified gravimetrically following Canfield et al., (1986): Canfield, D.E., Raiswell, R., Westrich, J.T., Reaves, C.M., Berner, R.A. 1986. The use of chromium reduction in the analysis of reduced inorganic sulfur in sediments and shales. Chem. Geol. 54, 149-155.

[Figure]

However, in agreement with some of the comments of Reviewer 2 we have now removed those portions of the manuscript which envisaged in situ production of tetrathionate/thiosulfate from pyrite; accordingly, methods concerning the estimation of pyrite have also been removed from the revised text.

Referee's Comment: Line 261: 0% partial pressure? Pressure unit is not percent. Also 0 probably means anoxic?

Authors' Response: Yes, 0% partial pressure means anoxic; the workstation was set at this specification following the manufacturer's instruction.

Referee's Comment: Line 268ff: This is the standard cline protocol, which is widely used and generally accepted - not necessary to describe the principle.

Authors' Response: We agree, and have now removed the mention of the principle.

Referee's Comment: Line 286: what does serially diluted mean?

Authors' Response: Serial dilution as a basic microbiological practice is very much similar to that in chemistry. Here a microbial inoculum (cell suspension) is sequentially diluted by orders of magnitude so as to reduce the density of cells in the suspension to such levels which give rise to manually-countable colonies when the suspension is spread out on solid media plates.

Referee's Comment: Line 288: here and elsewhere, please use until instead of till. As this is a scientific article, and till is considered to be informal which should be avoided.

Authors' Response: We agree and have now done so.

Referee's Comment: Line 289: "pure-plates"?

Authors' Response: Pure-plates refer to those microbial culture plates which have only one type of colonies, evidently representing one type of bacterium.

Results and discussion

Referee's Comment: Line 327 "relevant (microorganisms)" ?

Authors' Response: The sentence here reads as "Tetrathionate-forming, oxidizing, or respiring genes and relevant microorganisms are abundant in ..."; so it is presumable that the microorganisms, like the genes identified, also refer to tetrathionate forming/oxidizing/reducing entities.

Referee's Comment: Line 329-334: Very long sentence- almost not understandable: consider splitting and rewriting.

Authors' Response: The entire paragraph spanning between the previous line numbers 329-343 have now been simplified; as a part of this restructuring, the sentence in question has also been split into two.

Referee's Comment: Line 334 "were found to contain" – shorter "contained"

Authors' Response: Done, as suggested.

Referee's Comment: Line 341: Unpublished data should not be cited if not necessary. Here are 5 other references given. The reference to unpublished data is unnecessary.

Authors' Response: Done as suggested.

Referee's Comment: Line 391-395: Long and unclear sentence.

Authors' Response: We agree, and have now rewritten this sentence in a lucid way.

Referee's Comment: Line 406 – 411: Long and unclear sentence.

Authors' Response: We agree, and have now rewritten this sentence in a lucid way.

Referee's Comment: Line 423: The discussion refers here to another unpublished paper. The suggestion/discussion here is based on unpublished data from the authors. Such data should either be included in this manuscript or published first. Alternatively, the results should be discussed in the light of other already published studies. Otherwise, this discussion is not solid.

Authors' Response: For these data illustrating the feasibility of aerobic metabolism in these sediment horizons, please note that the same constitute a completely separate paper of our group, under consideration elsewhere, and those data are too voluminous to be incorporated here. Anyway, we have added in this revised manuscript that genes for aerobic respiration by aa3-/cbb3-type cytochrome-c oxidases (coxABCD / ccoNOPQ) and cytochrome-bd ubiquinol oxidase (cydABX / appX) were identified in the assembled metatranscriptome from 275 cmbsf of SSK42/6 in general, and the portions of the metatranscriptomic dataset in particular which matched with sequences from the tetrathionate-oxidizing isolates, thereby suggesting that potential activity of this aerobic metabolic process is possible in this environment (the relevant gene and transcript Tables have also been incorporated in the revised manuscript as Supplementary Materials).

Referee's Comment: Lines 425 ff. The results of the incubation appear very unsystematic or random. A figure could help for an overview. The writing is also not precise, i.e. the "samples" do not convert thiosulfate to tetrathionate : : : conversion was observed in the samples or the organisms in the sample convert the species: : :

Authors' Response: We have now made the presentation of all these results more systematic by adding new Figures and amending the language in the text in such a way that it is clear to the reader that the organisms in the samples converted the sulfur species. Furthermore, we have also amended the corresponding Methods section to put the principles of the slurry incubation experiments and the subsequent data in proper context.

Referee's Comment: Line 430ff: "In contrast, : : :" The sentence is very long. Also, there is no "contrast" obvious. "free and detectable" is unnecessary.

Authors' Response: We agree, and have now simplified this sentence as well as the entire text in this Results section; accordingly, the phrase "free and detectable" has been done away with. In this context, however, it is noteworthy that the data presented

in the lines 430ff regarding the thiosulfate oxidation property of the 45, 60 and 295 cmbsf communities of SSK42/5 are distinct from those concerning 0, 15, 90 and 160 cmbsf presented just in the previous sentence; hence the succeeding sentence ought to start with "In contrast".

Referee's Comment: Lines 434ff: the samples do not metabolize. Organisms have a metabolism but not a sediment samples. In this entire section is not clear how the rates were determined. A figure might help

Authors' Response: We agree, and have now replaced the word "metabolize" by the word "oxidize" in the context of this sentence. In addition, a number of lines within this section have been restructured, while a more elaborate description of the rationale behind the determination of rates has been provided. The entire set of data has also been presented graphically.

---

## Author Comment (AC3) · 20 Sep 2019

Referee's Comment: The authors investigate the capacity of microorganisms in sediments from the Arabian Sea to metabolize redox-intermediate sulfur species. This is an important question with significance for understanding carbon turnover in sediments as well as for interpreting sedimentary records, in which these rapid processes are often invisible. The study relies primarily on the successful culturing of organisms with at least the facultative capacity to metabolize tetrathionate, and on metagenomic and metatranscriptomics datasets for a series of depths in two cores. The data are

interesting and would fill an important gap in the literature, if the authors can address possible methodological concerns below. A more serious issue, however, appears in the argument regarding tetrathionate production from pyrite. This idea is not present in any of the three papers cited, which evidences a serious misunderstanding about this process and causes me to recommend rejection of the paper in its current form. Additional comments and suggestions follow.

{Note – I am not able to speak to the appropriateness of the metagenomic methods described here, since this is outside my area of expertise.}

Authors' Response: We thank the Reviewer for these comments, and believe he appreciated the underlying science of this study. We also agree with most of the concerns pointed out subsequently, so have now dealt with each one of them in course of this extensive revision.

So far as the citation oversight is concerned we have already tendered our apologies and explanations for the same during the open review process Furthermore, it has also been conveyed through those series of communications that we have now removed all those portions of the manuscript which discussed the potential pyrite-derived and MnO2-driven tetrathionate production in marine sediment (as the data in hand were not fully adept in handling that issue) and based all the discourse on origin of tetrathionate on the sound microbiological data. The relevant Specific Comments of the Referee (with respect to Line 492, and then Line 498), Authors' Initial Responses, Reviewer's Subsequent Comments, and finally Authors' Subsequent Response have all been appended below for your kind perusal.

Referee's Comment: General comment – for reading clarity, there are many places were codes could be simplified. You only have two cores – they could be referenced as A and B, or by collection depth, much more readably than SSK42/5 and SSK42/6. The names of slurry media types are similarly difficult to read (e.g. was 'T' tetrathionate or thiosulfate?).

Authors' Response: We agree that simpler nomenclatural codes could have been formulated for the sediment-cores. However, the purpose of adhering to the present nomenclature based on the cruise identity (RV Sindhu Sankalp, SSK42) is to maintain referral consistency across the series of papers we are currently publishing based on various biogeochemical aspects of the ten different cores collected on board SSK42, and named SSK42/1 through SSK42/10. For instance, the same SSK42/5 and SSK42/6 have been analyzed (alongside SSK42/1 through 4, SSK42/7 and SSK42/8) from separate perspective in Fernandes et al., 2018; so we believe that the readers should be allowed to clearly identify that these are the same 5th and the 6th cores of the SSK42 cruise that were also investigated and reported in Fernandes et al., 2018.

As for the nomenclature of the media types, please note that throughout the manuscript, codes for thiosulfate-containing media ended with 'T' while those for tetrathionate-containing media ended with 'Tr' (and not 'T'). However, in the revised manuscript, we have now incorporated further clarification for the basis of this nomenclature by stating as follows. "For each experiment testing the formation or oxidation of tetrathionate, 10% (w/v) sediment-sample was suspended in artificial sea water (ASW) supplemented with thiosulfate (T) or tetrathionate (Tr), i.e. ASWT or ASWTr broth medium (Alam et al., 2013), respectively; the culture flask was incubated aerobically at 15°C on a rotary shaker (150 rpm)."

Referee's Comment: Bulk sediment porewater geochemistry – please consider converting information in Table 1 to a pair of depth profile figures, one for each site. Please provide an appropriate number of significant figures for your data (you are currently reporting sub-nM precision).

Authors' Response: We agree, and have now included depth profile figures in the revised manuscript and also provided an appropriate number of significant figures for the data (notably, the current manuscript does not report any concentration in sub-nM precision). As for significance of values, in the revised manuscript, we have now mentioned (i) the different concentrations of sulfate, sulfide and thiosulfate that were

measured for the construction of the corresponding calibration curves, and (ii) what maximum percentage of deviation was found from actual concentrations for sulfate, sulfide or thiosulfate based on triplicate analyses of the respective sets of standards.

4. Thiosulfate and sulfide concentrations – No details are provided about how porewater sampling was conducted, or how porewater samples were chemically preserved. (Were zinc acetate, bromobimane, or other preservatives used? How much time elapsed between collection and IC analysis? What transformations might have occurred among your redox-sensitive dissolved species?)

Authors' Response: We have now overhauled the portion dealing with sampling details (including those for pore-waters) in such a way that all confusions emanating from the previous text are now resolved. This said, it is noteworthy that answers to these questions were already there in our previous publication [Fernandes et al., 2018, Enhanced carbon-sulfur cycling in the sediments of Arabian Sea oxygen minimum zone center. Sci. Rep. 8: 8665] which also dealt with these SSK42 cores (albeit form other perspectives) and was cited a number of times within the sampling-related section of the present manuscript. We had therefore thought that further repetition of the details would be redundant. However, now we understand that as an independent paper this manuscript should carry its own sampling details and have therefore brought back the necessary details taking sufficient care of literary repetitiveness; for instance, we have now mentioned in this revised manuscript also that (i) sodium azide was used to arrest further microbial activity within the samples for chemistry, (ii) all IC analyses were completed within one week of retrieval of the samples to the laboratory on land, and (iii) all the pore-water vials were crimped immediately after N2 flushing, and stored at 4 °C until further analysis.

Referee's Comment: Depth trends - Figure 1 seems to show that all five categories of relevant genera increase in their relative proportion smoothly with depth, which seems like it could be to first order a change in dilution of the microbial population with organisms that prefer the shallower sediments (aerobes or those that subsist on fresh,

relatively shallow organic matter). The significance of this first-order depth trend needs to be discussed separately in the discussion. Standard depth profiles of the five genera groups would be much easier for the reader to interpret.

Authors' Response: We completely agree with your appreciation of the depth trends, so in the revised version of the manuscript we have now discussed all these issues under a new section titled "Trends of geomicrobial parameters down the sediment-depths corroborated sulfur cycle functions centered on tetrathionate".

Referee's Comment: A more thorough description of how cores 42/5 and 42/6 differ sedimentologically would improve the analysis, especially since the key correlations from Figs 1 and 2 are only strongly significant in one of the cases.

Authors' Response: In the revised manuscript we have now appended a thorough explanation as to why the key correlation coefficients such as those indicated in Figs 1 and 2 are only strongly significant in case of SSK42/5 and slightly weaker for SSK42/6. As you would see discussed in the revised manuscript, relative abundances of metagenomic reads ascribed to the genera of tetrathionate-forming, oxidizing, and respiring bacteria also fluctuate more or less synchronously along SSK42/6, excepting the region between 250 and 275 cmbsf (Fig. 2), which is the sulfate-methane transition zone (SMTZ) of this sediment-horizon (notably, SMTZ in the SSK42/5 sediment-horizon laid below the 280 cmbsf sediment-depth explored in this core; see Fernandes et al., 2018 for the methane profiles of the SSK42 cores). Whilst lack of synchrony in the lower end of this core resulted in relatively weaker correlation values as compared to those obtained for SSK42/5 (Table S7), we hypothesize that the changes in geochemistry and community architecture associated with the advent of SMTZ potentially impact the in situ population ecology of the tetrathionate-metabolizing bacterial types also. Sedimentation rate, age-depth profile and other geochemical features of the two cores separated by a distance of only one kilometer are otherwise largely comparable.

Referee's Comment: Line 267 – I read this statement to mean that some representatives of each of these groups have been observed to cycle tetrathionate. Clearly, though, that does not apply to all members of these broad taxonomic groups, which makes it very difficult to tell whether the organisms might generally or facultatively cycle tetrathionate or something else entirely. Are statistics available on what fraction of, say, Salmonella the genetic machinery for tetrathionate conversion has?

Authors' Response: Line 267 did not deal with the kind of issues that you have mentioned here as it actually encompassed the Materials and Methods section. However we presume that your concerns are centered on the articulations that were made in lines 372-411. There, it has been clearly distinguished (already in the initial manuscript) that for one category of genera each and every member strain in the literature is known to possess tetrathionate-forming/oxidizing property, so the presence of such genera is more definitely indicative of the concerned processes in situ, whereas for another category of genera only some (and not all) member strains are known to possess tetrathionate-forming/oxidizing/reducing property, so their presence indicates further additional possibilities of such processes in situ. Furthermore, it may be noted that to keep this discrimination explicit, trends of relative abundance for the first category were depicted in Figs. 1 and 2 (these data are clearly free from diversity/abundance over-estimation), while those for the second category were all presented separately in Supplementary Tables S8-S13 (these data are likely to involve unknown proportions of diversity/abundance over-estimation, and so have been kept in isolation from the definitive estimates given in Figs. 1 and 2).

We have now edited the text in such a way as to make these issues more clearly comprehendible.

Referee's Comment: Line 423 –Unless the data is included here, your own unpublished conclusions can't be cited like this.

Authors' Response: For these data illustrating the feasibility of aerobic metabolism in these sediment horizons, please note that the same constitute a completely separate

paper of our group, under consideration elsewhere, and those data are too voluminous to be incorporated here. Anyway, we have added in this revised manuscript that genes for aerobic respiration by aa3-/cbb3-type cytochrome-c oxidases (coxABCD / ccoNOPQ) and cytochrome-bd ubiquinol oxidase (cydABX / appX) were identified in the assembled metatranscriptome from 275 cmbsf of SSK42/6 in general, and the portions of the metatranscriptomic dataset in particular which matched with sequences from the tetrathionate-oxidizing isolates, thereby suggesting that potential activity of this aerobic metabolic process is possible in this environment (the relevant gene and transcript Tables have also been incorporated in the revised manuscript as Supplementary Materials).

Referee's Comment: Much of section 3.3 is a reporting of results without context or discussion, which is challenging to parse for key points – consider separating out your results. Figures would be very helpful. How do these rates compare with other culture studies or with the size of the porewater pools? What are these depth patterns? What do you want your reader to gain from this information?

Authors' Response: We agree that discussion of the slurry incubation data in the context of the rest of the results would enrich the manuscript, and so have done the same under a new Discussion section 4.1 titled "Trends of geomicrobial parameters down the sediment-depths corroborated sulfur cycle functions centered on tetrathionate". Furthermore, as per your suggestions, we have now incorporated new figures to depict the depth trends of the slurry incubation data. This said, it may be noted that the main purpose of the slurry incubation experiments were to check whether the tetrathionate-metabolizing bacteria are alive in situ. Accordingly, the two Result sections 3.3 and 3.4 were (and are still) titled as - The tetrathionate-forming/oxidizing microorganisms of the ASOMZ sediments are alive and active in situ, and - Active tetrathionate-reducing microorganisms in ASOMZ sediment, respectively. Notably further, within 3.3 and 3.4, the "live and active" issue was addressed by pure-culture isolations and metatranscriptomics in addition to the slurry incubation experiments.

Referee's Comment: The title of section 3.3 isn't quite true – these experiments show the presence of living organisms that can at least facultatively do these metabolisms; it does not show that they are actively conducting these metabolisms in-situ. Can you integrate this conclusion with your RNA results or show actual in-situ abundances of your cultured organisms?

Authors' Response: We thank you for this nice suggestion and believe the kind of data recommended would go a long way in making the conclusions robust. Accordingly, we have carried out whole genome shotgun sequencing and annotation for the three tetrathionate-forming isolates Halomonas sp. MCC 3301, Methylophaga sp. MTCC 12599 and Pseudomonas bauzanensis MTCC 12600; the two tetrathionate-oxidizing isolates Halothiobacillus sp. SB14A, and Pusillimonas ginsengisoli MTCC12558; and the tetrathionate-reducing isolate Enterobacter sp. RVSM5a. Subsequently we have mapped the metagenomic sequence data from the 25 distinct sample-sites of SSK42/5 and SSK42/6 separately onto each of above mentioned de novo sequenced genomes – remarkably, significant percentages of the metagenomic read-sets were found in this way to match sequences from the individual genomes. The data, which clearly give a picture of the relative abundances of the strains in each of the 25 distinct sediment-samples have been presented in the form of a new heat map figure.

Referee's Comment: Line 385 – intracellular vs extracellular tetrathionate. Most of these examples of tetrathionate producers generate tetrathionate as an intracellular intermediate species during metabolism. Please discuss how you envision other members of the microbial community accessing these species.

Authors' Response: We agree that literature is very scant and obscure (limited to only two papers of 1989 and 1992) in relation to tetrathionate formation during sulfate/sulfite reduction by members of the genera Desulfovibrio and Desulfobulbus (which were mentioned in Line 385 of the previous manuscript). Whilst in the two papers available there is no clear-cut indication of whether the tetrathionate formed during sulfate/sulfite reduction by these organisms appear in the spent-media or not, the amount

of tetrathionate produced in those cases is said to be in the micro molar range, so intracellular accumulation, if that is at all the case, would not pose major physiological problem for the cells. In view of these uncertainties in the knowledge base we have now removed these two instances from the list of bacteria cited as additional and likely sources of tetrathionate, both from the text as well as Tables S8 and S9.

Notably, however, unlike for the two sulfate-reducers mentioned above, there are definite reports from sulfur-oxidizing chemo-/photo-lithotrophic bacteria that polythionates (including tetrathionate) – whether provided in the media as substrates or formed during the oxidation of thiosulfate - do not accumulate intracellularly (only a few of the several relevant references are given below). All polythionates have high biological reactivity, and the concentrations of polythionates involved in lithotrophic growth of bacteria are in the milli molar range, so most of the chemolithotrophic bacteria are known to uptake and use polythionates as and when required but never accumulate the same intracellularly. Moreover, the papers cited below have amply demonstrated in pure culture experiments that polythionate species formed from thiosulfate, whether subsequently oxidized to sulfate or not, are released into the extracellular milieu (spent media) as detectable (by cyanolysis method) solutes.

Given this physiological aspect of lithotrophic tetrathionate production, any other tetrathionate-metabolizing bacteria present in the system, whether a natural environment or a mixed culture, would get equal opportunity to utilize the polythionate formed as the former organism itself gets.

1. Ghosh, W., Bagchi, A., Mandal, S., Dam, B. and Roy, P., 2005. Tetrathiobacter kashmirensis gen. nov., sp. nov., a novel mesophilic, neutrophilic, tetrathionate-oxidizing, facultatively chemolithotrophic betaproteobacterium isolated from soil from a temperate orchard in Jammu and Kashmir, India. International Journal of Systematic and Evolutionary Microbiology 55: 1779–1787. 2. Wood, A.P. and Kelly, D.P., 1985. Physiological characteristics of a new thermophilic obligately chemolithotrophic Thiobacillus species, Thiobacillus tepidarius. International Journal of Systematic and Evolutionary

Microbiology 35: 434-437. 3. Wood, A.P., Woodall, C.A. and Kelly, D.P., 2005. Halothiobacillus neapolitanus strain OSWA isolated from "The old sulphur well" at Harrogate (Yorkshire, England). Systematic and Applied Microbiology 28: 746-748. 4. Hensen D., Sperling D., Truper H.G., Brune D.C. and Dahl C., 2006. Thiosulphate oxidation in the phototrophic sulphur bacterium Allochromatium vinosum. Molecular Microbiology 62: 794–810. 5. Alam, M., Pyne, P., Mazumdar, A., Peketi, A. and Ghosh, W., 2013. Kinetic enrichment of 34S during proteobacterial thiosulfate oxidation and the conserved role of SoxB in S-S bond breaking. Applied and Environmental Microbiology 79: 4455-4464. 6. Pyne, P., Alam, M., Rameez, M.J., Mandal, S., Sar, A., Mondal, N., Debnath, U., Mathew, B., Misra, A.K., Mandal, A.K. and Ghosh, W., 2018. Homologs from sulfur oxidation (Sox) and methanol dehydrogenation (Xox) enzyme systems collaborate to give rise to a novel pathway of chemolithotrophic tetrathionate oxidation. Molecular Microbiology 109: 169-191.

Referee's Comment: Line 445-453 - How do you possibly get as many as six separate samples that have identical observed rates (e.g., of 141ï′Cs′ 1 ïAËŻ mol S/d/g)?

Authors' Response: In the above experiments it was remarkable that the individual communities present within the sediment-depth-zones spanning 2-90 cmbsf, 120-175 cmbsf, or 220-275 cmbsf, exhibited mutually identical rates of tetrathionate oxidation despite having dissimilar composition/prevalence of chemolithotrophic taxa. This could be explained as follows. When a natural sample is incubated in selective culture media (such as ASWTr) certain specific microbial species present in the sample often outgrow all metabolic competitors by virtue of higher substrate affinity or culture-condition suitability. Consequently, the growth/substrate-utilization phenotype(s) manifested by such enriched consortium cultures are contributed to by the selected few rather than the entire community of metabolic equivalents present in the sample (Roy et al., 2016). In the light of these issues it seems quite plausible that specific sets of chemolithotrophic taxa, more adept to ASWTr-growth than others, are present across the sediment-samples within the 2-90 / 120-175 / 220-275 cmbsf zones, and it was only their typical rates of

tetrathionate oxidation in vitro which we incidentally recorded as the in vitro tetrathionate oxidation rates of the communities. Whatever may be the actual tetrathionate formation/oxidation rate of the SSK42 sediment-samples in vitro or in situ, results of the slurry culture experiments illustrated that tetrathionate-forming and oxidizing bacteria of SSK42/5 and SSK42/6 were alive in situ.

Referee's Initial Comment: Line 492 – The authors cite three papers to support a link between MnO2 cycling and pyrite oxidation to tetrathionate and other dissolved species. I have no idea what the authors are referencing, which is troubling and forces me to recommend rejection. The Berner and Petsch paper from 1998 does not include the words manganese, thiosulfate, or tetrathionate. There is similarly no mention of MnO2 in the Luther 1991 paper. And, although the Jørgensen and Bak paper discusses manganese, it is in the context of "manganese or iron oxides" which could similarly be used as electron acceptors, not anything about pyrite oxidation.

Authors' Initial Response: WE ARE EXTREMELY SORRY for this "copy-paste" goof-up committed in the haste of submitting multiple manuscripts within the same time-window! The actual reference that should have been used is Schippers, A., and Jørgensen, B. B.: Oxidation of pyrite and iron sulfide by manganese dioxide in marine sediments, Geochim. Cosmochim. Acta., 65, 915-922, https://doi.org/10.1016/S0016-7037(00)00589-5, 2001. (Please note that this was already included in the Reference list and cited in another context of Discussion). This paper should have also been cited in this particular context of tetrathionate production from pyrite, instead of the three irrelevant references that mistakenly got inserted. Please see Line numbers 4-8 of the Abstract itself of Schippers and Jørgensen, 2001, which clearly states that "FeS2 and iron sulfide (FeS) were oxidized chemically at pH 8 by MnO2 but not by nitrate or amorphic Fe(III) oxide. Elemental sulfur and sulfate were the only products of FeS oxidation, whereas FeS2 was oxidized to a variety of sulfur compounds, mainly sulfate plus intermediates such as thiosulfate, trithionate, tetrathionate, and pentathionate. Thiosulfate was oxidized by MnO2 to tetrathionate while other intermediates were oxidized to sulfate."

Referee's Subsequent Comment: Thank you for clarifying the correct references here, your thinking is far clearer now. Although this reference does describe pyrite oxidation via MnO2, I am still unconvinced that it is relevant to the sediments in the current study. If one reads beyond the abstract of that paper, one also finds that "Below 7.5 cm, where the content of Mn did not exceed 0.2% (w/w), a dissolution of FeS2 was not detectable." One also finds that the abiotic incubations produced intermediately only for days to weeks and not longer, decreasing rapidly with depth. Manganese concentrations in the current paper (71-172 ppm) are orders of magnitude lower than the threshold for activity reported before, and there are no depth trends in either pyrite or MnO2 discussed, or porewater metal ion data, that might support this mechanism as active. Purported Mn driven oxidation also appears to increase, rather than decrease, with depth, in contrast with the prior report. I do not think pyrite is a source of dissolved S species in this system; stronger evidence is required.

Authors' Subsequent Response: We agree with your observation that the MnO2-FeS2 interaction as a source of tetrathionate has been overstretched and indeed needs more experimentation to substantiate. We agree to curtail this particular discussion component and limit only to the observation and conclusion drawn from the microbial studies.

Referee's Initial Comment: Line 498 – The entire argument for pyrite oxidation by MnO2 appears to be that there is detectable Mn in the sediments. (Basically all sediments have this??) There must be more one could say on this topic : : : depth trends? Differences between the two cores? Comparison with typical sediment Mn concentrations? Otherwise I'd leave the Mn discussion out. I would certainly not use this discussion to conclude that "Pyrites (via abiotic reaction with MnO2) and thiosulfate (via chemolithotrophic oxidation by members of the bacterial group designated as A in Fig. 4) are apparently the main sources of tetrathionate". This has not been demonstrated.

Authors' Initial Response: Chemolithotrophic conversion of thiosulfate to tetrathionate

by members of the bacterial genera Pseudomonas and Halomonas (designated as A in Fig. 4) has been experimentally demonstrated - we have shown such isolates of both Pseudomonas and Halomonas which are capable of chemolithotrophically converting thiosulfate to tetrathionate in vitro. Furthermore, when metagenomic sequence data obtained for each of the 25 distinct sediment-samples of SSK42/5 and 6 were assembled and annotated individually, 23 out of the 25 contig-collections obtained were found to contain genes for tetrathionate formation (namely, genes encoding subunits of the thiosulfate dehydrogenases TsdA that converts thiosulfate to tetrathionate; see Denkmann et al., 2012; Pyne et al., 2018) [Table S3]. Whole metatranscriptome sequencing and analysis for the 275 cmbsf sediment-sample of SSK42/6 also revealed the gene-catalog obtained via annotation of the assembled contigs to encompass homologs of the thiosulfate dehydrogenase gene tsdA [Table S19]. These data clearly supported the potential in situ functionality (metabolically active state) of thiosulfate to tetrathionate converting bacteria.

Referee's Subsequent Comment: I do not dispute that thiosulfate-to-tetrathionate conversion was demonstrated and is quite intriguing; the piece of your claim that has not been demonstrated is related to pyrite. Without showing any depth trends, porewater metal ion data, or pyrite-specific (tracer) incubations, there is no data evidencing the involvement of pyrite.

Authors' Subsequent Response: We agree with your observation that the MnO2-FeS2 interaction as a source of tetrathionate has been overstretched and indeed needs more experimentation to substantiate. We agree to curtail this particular discussion component and limit only to the observation and conclusion drawn from the microbial studies.

---

## Author Comment (AC4) · 20 Sep 2019

Referee's Comment: The manuscript describes the analysis of inorganic sulfur compound cycling microbial populations in two sediment cores from the Indian Ocean. Particular focus is on populations driving the metabolism of thiosulfate and tetrathionate (thiosulfate reducing/tetrathionate forming, tetrathionate reducing, and tetrathionate oxidising groups). The study used a range of geochemical measurements, slurry incubations, microbiology (isolation of sulfur cycling microorganisms and assessment of their capabilities to transform thiosulfate and tetrathionate) as well as molecular biological approaches (metagenomics and metatranscriptomics). The biogeochemistry of sulfur compounds in sediments is a complex web of chemical and biological transformations, there is a need to better understand the role of individual species of inorganic sulfur compounds as well as the metabolic pathways and microbial groups involved in their transformations. As such, this is a topic of high interest, especially if, as the title implies, some of the transformations may be of a cryptic (not easily identified) nature.

Authors' Response: We thank the Reviewer for appreciating the underlying science of this study.

Referee's Comment: My overarching impression of the manuscript is that it is not easy to follow the story and that it would benefit from revising the structure. It is lacking a clear approach to the analysis and presentation of the data. Even starting in the introduction, I would suggest that, given the focus on the various enzymes being instrumental in the transformations of thiosulfate and tetrathionate, the introduction should provide a brief overview of the most important enzymes involved (and their encoding genes) and perhaps contain a schematic conceptual overview illustrating the most important points.

Authors' Response: We agree with your concerns, and so in the revised version of the manuscript have overhauled the entire text by providing the warranted information (regarding the enzymes and genes which appear to be instrumental in the transformations of thiosulfate and tetrathionate) in the Introduction, making the rationales for the analyses clearer, and discussing the results in their proper context under separate Results / Discussions sections. A schematic conceptual overview illustrating the most important points was already there in the form of Figure 4 in the previous manuscript and the same has now been updated with respect to the altered data components of the paper.

Referee's Comment: It would be beneficial and aid readability, if a clear overview of the basic findings was shown perhaps as depth profiles showing key chemical parameters

of the cores under investigation.

Authors' Response: We agree, and have now incorporated new figures showing depth profiles of all the important geochemical and microbiological parameters and comprehensively discussed their implications under a new Discussion section 4.1 titled "Trends of geomicrobial parameters down the sediment-depths corroborated sulfur cycle functions centered on tetrathionate".

Referee's Comment: With a view of the diversity and metagenomics analysis, I have two key criticisms: (i) revolving around the specific use of metagenomics read data for taxonomic assignment and (ii) extrapolating from that assignment to physiological properties of entire genera of bacteria. In that context, I have to say that I think it is a pity the authors did not carry out a diversity analysis of the sediment samples based on pooled 16S rRNA amplicon sequencing in parallel to the metagenomics/metatranscriptomics, because the ribosomal RNA gene survey would provide a much better and more robust diversity analysis than the assignment of taxonomy based on random metagenomic reads. Although a taxonomic assignment of a random metagenomic read is possible, it is fraught with major uncertainty, unless a closely related organism's genome is available in a database. As that is not the case for the vast majority of microorganisms found in nature at present, the taxonomic assignment of metagenomic reads is a bound to provide unreliable/unresolved taxonomies and lead to poor estimates of the abundance of specific types of bacteria. This affects data shown in Tables S8-13 as well as Fig 1 and 2.

Authors' Response: We agree that rRNA gene sequence analyses provide more robust taxonomic diversity analysis than the assignment of taxonomy based on identity of shotgun metagenomic reads; but on the flip side we have to bear in mind that such data only remain qualitative and give estimates of alpha diversity. We have already published the comprehensive, 16S rRNA gene sequence based, taxonomic diversity analyses for almost 100 sediment-samples from six SSK42 cores, including SSK42/5 and SSK42/6 [Fernandes et al., 2018, Enhanced carbon-sulfur cycling in the sediments of Arabian Sea oxygen minimum zone center. Sci. Rep. 8: 8665]. But such PCR amplified sequence data are not theoretically suitable for interpreting any quantitative population ecology trends or community metabolic functions such as the ones we have revealed in this paper from shotgun metagenomic sequence datasets (by directly annotating the same via BLAST search against the non-redundant protein sequence database, or assembling and annotating genes within the contigs by HMMER search against the EggNOG database).

We also agree with your concerns that taxonomic assignment from shotgun metagenomic reads, though possible, is fraught with major uncertainty, unless a closely related organism's genome is available in a database. With regard to this issue we can assure you that the parameters we have used to classify reads using the Best Hit Classification algorithm [BlastX search with minimum 45 nucleotides (15 amino acids) alignment and $\geq$60% identity, and maximum e-value allowed 1e–5] are stringent enough to assign taxonomic affiliation to homologs of metabolically diverse genes, irrespective of their intrinsic levels of conservation, in a reliable manner up to the genus level. This stringency level of search parameters is considered optimum across the literature because it neither exaggerates diversity not fails to resolve taxonomies for most categories of genes.

Anyway, there is no doubt that these are virtually never-dying debates, so to lay all apprehensions to rest we have carried out whole genome shotgun sequencing and annotation for the three tetrathionate-forming isolates Halomonas sp. MCC 3301, Methylophaga sp. MTCC 12599 and Pseudomonas bauzanensis MTCC 12600; the two tetrathionate-oxidizing isolates Halothiobacillus sp. SB14A, and Pusillimonas ginsengisoli MTCC12558; and the tetrathionate-reducing isolate Enterobacter sp. RVSM5a. Subsequently we have mapped the metagenomic sequence data from the 25 distinct sample-sites of SSK42/5 and SSK42/6 separately onto each of above mentioned de novo sequenced genomes – remarkably, significant percentages of the metagenomic read-sets were found in this way to match sequences from the individual genomes.

The data, which clearly give a picture of the relative abundances of the strains in each of the 25 distinct sediment-samples have been presented in the form of a new heat map figure.

Referee's Comment: My second criticism is the assumption in the paper that entire genera of bacteria always share specific physiological capacities with respect to the sulfur transformations of interest. While this is true for some genera (and a parameter used in systematics), for many genera this is not necessarily the case. Therefore, suggesting that genus A or B are tetrathionate producing bacteria or -oxidising bacteria, will likely overestimate the abundance of that specific metabolic type based on that assumption.

Authors' Response: So for as direct taxonomic annotations of raw metagenomic reads, followed by functional/metabolic classification of taxa, are concerned please note that we have all along clarified (already in the initial manuscript) that for one category of genera each and every member strain in the literature is known to possess tetrathionate-forming/oxidizing/reducing property, so the presence of such genera is more definitely indicative of the concerned processes in situ, whereas for another category of genera only some (and not all) member strains are known to possess tetrathionate-forming/oxidizing/reducing property, so their presence indicates further additional possibilities of such processes in situ. Furthermore, it may be noted that to keep this discrimination explicit, trends of relative abundance for the first category were depicted in Figs. 1 and 2 (these data are clearly free from diversity/abundance over-estimation), while those for the second category were all presented separately in Supplementary Tables S8-S13 (these data are likely to involve unknown proportions of diversity/abundance over-estimation, and so have been kept in isolation from the definitive estimates given in Figs. 1 and 2).

We have now edited the text in such a way as to make these issues more clearly comprehendible.

[Figure]

Referee's Comment: On the other hand, the metagenomics data can reveal the abundance of specific types of genes, as has been done here, and potentially identify the types of bacteria potentially contributing to the cycling of specific compounds. Too much of the discussion of inorganic sulfur metabolism in these sediments is based on the broad assumption of taxonomy, and too little is made of the specific genes found and listed in various supplementary tables. There are still limitations of our understanding of the genetics of sulfur transformations and it would be useful to perhaps illustrate whether the enzymes/genes driving specific transformations of sulfur compounds in some of the taxa mentioned (eg Salmonella) have actually been identified. Very little is done with the metatranscriptome data, it only gets a few mentions, but there is no clear overview of what has been found in which layer, how many reads were analysed and generally which bacteria were transcriptionally active with respect to sulfur cycling.

Authors' Response: We absolutely agree that there are major limitations in global understanding of the genetics of sulfur transformations; therefore, we have now reviewed in the new Introduction all the enzymes/genes that have been identified thus far as drivers of specific transformations of sulfur compounds in the taxa mentioned/considered in the text (including Salmonella). In this context it is noteworthy that corroborating your assumption, the genome of the current tetrathionate-reducing isolate belonging to the genus Enterobacter was found not to encompass the typical tetrathionate reductase (ttr) genes. For the records, we have now also added a detailed analysis of the six new genomes in relation to the homologs of sulfur-transformation genes present.

As for the metatranscriptome analyses, we have now added new data identifying which tetrathionate-metabolizing genes in general, and those matching homologs from the genomes of the new isolates in particular, were there in the metatranscriptome. In the revised manuscript we have also included dedicated sections for in depth metatranscriptomic methodology where the all the read statistics have been given. Notably, however, metatranscriptome was analyzed for only 275 cmbsf of SSK42/6, so the data

obtained thereof are not applicable to the other layers of the two sediment packages.

Referee's Comment: Regarding the transformations measured in slurry experiments, there needs to be a more complete reporting of the activities measured (or not) in all sediment samples. This should be shown comprehensively, not as currently done in Tables S14-S21, which suggest that only a few subsamples had certain activities. I am also not convinced that a 30-day incubation period of the slurries, some incubating sediments from a completely anoxic system under aerobic conditions (!), is providing the sort of activity data that would be supportive of suggesting that these key biological reactions are linked up in a cryptic cycle. The mentioning of alternative sources of oxygen from cryptic sources such as perchlorate is pointing to an unpublished study by the same authors. If this is crucial for the understanding of the functioning of this system and the microbiological activities required for the cryptic cycling of these compounds, these aspects should be incorporated here or the other study needs to be published. Alternatively, it is possible that the activities are due to facultative anaerobic bacteria in these sediments that have reverted back to an aerobic lifestyle given suitable incubation conditions and a 30-day period to wake up.

Authors' Response: We have now reported the slurry culture data as graphs plotted against sediment-depths. Notably, Whatever may be the actual tetrathionate formation/oxidation rate of the SSK42 sediment-samples in vitro or in situ, results of the slurry culture experiments illustrated that tetrathionate-forming and oxidizing bacteria of SSK42/5 and SSK42/6 were alive in situ. The issue of potential in situ active state of the tetrathionate-metabolizing bacteria was addressed mainly by the metatranscriptomic data both assembled and generally annotated as well as mapped against tetrathionate metabolizing genes of the isolates.

As for the data illustrating the feasibility of aerobic metabolism in these sediment horizons, please note that the same constitute a completely separate paper of our group, under consideration elsewhere, and those data are too voluminous to be incorporated here. Anyway, we have added in this revised manuscript that genes for

aerobic respiration by aa3-/cbb3-type cytochrome-c oxidases (coxABCD / ccoNOPQ) and cytochrome-bd ubiquinol oxidase (cydABX / appX) were identified in the assembled metatranscriptome from 275 cmbsf of SSK42/6 in general, and the portions of the metatranscriptomic dataset in particular which matched with sequences from the tetrathionate-oxidizing isolates, thereby suggesting that potential activity of this aerobic metabolic process is possible in this environment (the relevant gene and transcript Tables have also been incorporated in the revised manuscript as Supplementary Materials).

Specific comments Referee's Comment: Introduction: The introduction would benefit from a description of relevant metabolic pathways and enzymes targeted by the analysis of this paper Line 73: define mbsl Line 75 what was the diameter of these cores?

Authors' Response: We agree, and have now incorporated the warranted information in the Introduction. The diameter of all SSK42 cores was 12 cm; this information has been incorporated alongside the full-forms of the oceanographic units.

Referee's Comment: Line 82: for ease of reading I suggest to always refer to both cores with the full abbreviation, not SSK42/5 and 6 but SSK42/5 and SSK42/6

Authors' Response: We agree, and have now done as warranted.

Referee's Comment: Line 82: I find the description of the N2 shower lacking in detail and find it hard to understand how it would keep the exposed core adequately protected from oxygen.

Authors' Response: We have now overhauled the portion dealing with sampling details in such a way that all confusions emanating from the previous text are now resolved. This said, it is noteworthy that answers to these questions were already there in our previous publication Fernandes et al., 2018, which also dealt with these SSK42 cores (albeit form other perspectives) and was cited amply throughout the manuscript, including the sampling-related section. We had therefore thought that further repetition of the

details would be unnecessary and also potent causes of unintended self-plagiarism. However, now we understand that as an independent paper this manuscript should carry its own sampling details and have therefore brought back many of those details taking sufficient care of literary repetitiveness. For the records please note the following.

- In order to protect the ASOMZ sediment-samples from aerial oxidation, the entire cores were not split open into two D-shaped halves directly; instead only one ∼15-cm-long C-shaped part of PVC core-liner was removed at a time, as shown in a new Supplementary Figure added to the revised manuscript. We have now restructured the old sentences and written additional new lines to explain this in an unambiguous way. - The 15 cm length exposed at a time for sampling was constantly and closely showered with high-purity N2 emitted from multiple nozzles fitted to multiple nitrogen-generators. This contrivance was sufficient to prevent atmospheric oxidation of the 15 cm exposed surface of the core. - Immediately after the C-shaped longitudinal part of the PVC core-liner was cut open, top one cm of the exposed surface was scrapped off along the core-circumference using sterile scalpels to eliminate potential contaminations from the core-liners' inner-surfaces and/or sea-waters through which the cores had passed. - Subsequently, to sample a particular sediment-depth of the core for microbiological studies, an approximately 5-mm-thick sediment-slice (spanning equally on either side of the core-height marking) was scooped out with a sterile scalpel and put into a sterile polypropylene bottle. The head-space of every sample-containing bottle was flushed with high-pure N2, following which it was sealed with Parafilm - At the same time, for on board extraction of pore-waters, sediment-samples from a particular depth were taken out by inserting sterile 50 ml cut off syringes deep inside the core cross-section, multiple times along the circumference on the exposed 'C half'; the samples were immediately collected in sterile 50 ml centrifuge tubes; and all these were carried out under focused streams of high-pure N2.

Referee's Comment: Line 88: scraped

Authors' Response: We apologise, the typo is now fixed.

Referee's Comment: Line 145 define cmbsf, how thick was that sediment layer?

Authors' Response: cmbsf is now defined; approximately 5-mm-thick sediment-slices were sampled.

Referee's Comment: Line 147 and 190: pooling up, no 'up'

Authors' Response: We agree, now fixed.

Referee's Comment: Line 213: kmer lenghts of . . .

Authors' Response: Thank you, "lengths of" now added.

Referee's Comment: Line 227: text string search?

Authors' Response: Yes.

Referee's Comment: Line 236/7: not every read would represent one of these functional groups. Please reword/revise or explain more clearly what you did. There are no 'genera' of this that and the other, they are genera that contain species some of which have certain physiological characteristics, but not necessarily all of them

Authors' Response: We agree that your concerns hold well in a large number of cases, and exactly for that reason, we had clearly distinguished (already in the initial manuscript) that for one category of genera each and every member strain in the literature is known to possess tetrathionate-forming/oxidizing/reducing property, so the presence of such genera is more definitely indicative of the concerned processes in situ, whereas for another category of genera only some (and not all) member strains are known to possess tetrathionate-forming/oxidizing/reducing property, so their presence indicates further additional possibilities of such processes in situ. Furthermore, it may be noted that to keep this discrimination explicit, trends of relative abundance for the first category were depicted in Figs. 1 and 2 (these data are clearly free from diversity/abundance over-estimation), while those for the second category were all presented separately in Supplementary Tables S8-S13 (these data are likely to involve unknown proportions of diversity/abundance over-estimation, and so have been kept in isolation from the definitive estimates given in Figs. 1 and 2).

We have now edited the text in such a way as to make these issues more clearly comprehendible.

Referee's Comment: Line 246 following: No context for why one would assess the aerobic metabolism of these compounds with samples from 275cm below the sediment surface where there is no oxygen.

Authors' Response: Please note that since this study was aimed at revealing potential roles of tetrathionate in the sulfur cycle of marine sediments, plausible abilities to oxidize thiosulfate to tertahionate, and oxidize/reduce tetrathionate, were tested for all the communities present at all individual sediment-depths (from surface to core-bottom) of the sediment horizons explored and not just 275 cmbsf alone. Pure culture isolations and metatranscriptome analysis, however, were done only from 275 cmbsf, which is within the sulphate-methane transition zone where microbiological activity, for any marine sediment horizon, is generally very high and multi-faceted.

Referee's Comment: Line 309: actually described below

Authors' Response: We apologise, the oversight is now fixed.

Referee's Comment: Line 327: reword, genes do not have these activities, they encode enzymes that transform the compounds

Authors' Response: We agree, now fixed.

Referee's Comment: Line 350: I do not think that all of marine Pseudomonas and Halomonas do that

Authors' Response: We agree that not all but majority of the marine strains of Pseudomonas and Halomonas have the metabolic capacity of conversion of thiosulfate into

tetrathionate. Anyway, we have now overcome this small but important question mark by carrying out whole genome shotgun sequencing and annotation for the current three tetrathionate-forming isolates Halomonas sp. MCC 3301, Methylophaga sp. MTCC 12599 and Pseudomonas bauzanensis MTCC 12600, subsequent to which we have mapped the metagenomic sequence data from the 25 distinct sample-sites of SSK42/5 and SSK42/6 separately onto each of above mentioned de novo sequenced genomes – remarkably, significant percentages of the metagenomic read-sets were found in this way to match sequences from the individual genomes. The data, which clearly give a picture of the relative abundances of the strains in each of the 25 distinct sediment-samples have been presented in the form of a new heat map figure.

Referee's Comment: Table S1- Please clarify what is meant with 1st or 2nd sample fraction

Authors' Response: To sample a particular sediment-depth of a core for microbiological studies, an approximately 5-mm-thick sediment-slice (spanning equally on either side of the core-height marking) was scooped out with a sterile scalpel and put into a sterile polypropylene bottle.

For every sediment-depth, two such sample-replicates or slices - designated for duplicate metagenome (plus other metaomics) analyses - were collected (these were designated as sample replicates 1 and 2); a third slice was taken for all culture-dependent studies (notably, in the revised manuscript, the word "sample fraction" is now replaced by "sample replicate").

Referee's Comment: Table S4- Not a single DoxA encoding gene was identified in any of the samples. Should it perhaps not be listed in Table S4 accordingly? Please state what the significance of the yellow highlighting is in this and the other excel based supplementary tables

Authors' Response: Thanks for the suggestions. We have now removed the mention of doxA and soxD from Tables S3 and S4.

As the supplementary tables are large and each excel sheet contains large number of genes we have tried to assist visual cognition by alternately highlighting genes with similar metabolic function.

Referee's Comment: Table S6 Define what is meant with prevalence and how it is quantified Unclear what is meant with correlation of metabolic type with sediment depth, when depth is not quantified here

Authors' Response: To reduce the complexity of some of the sentences, the term "prevalence" has been used to denote relative abundance (detailed description for the estimation of relative abundance from metagenomic data, and its synonymy with prevalence, is mentioned in the Methods).

Please note that both the ∼3-m-long cores were explored biogeochemically at 15 to 30 cm intervals, which here is regarded as a sediment depth. In this way, we considered sediment-depth and tetrathionate metabolizing bacterial groups as two quantitative parameters and tested their interdependence by determining Pearson correlation coefficient (CC) and/or Spearman rank correlation coefficient (RCC).

Referee's Comment: Tables S8 to S11 should have totals for the abundance of all types per depth.

Authors' Response: We agree and have now given these totals.

---

## Author Comment (AC6) · 24 Sep 2019

General comments:

Referee's Comment: The paper investigated the role of tetrathionate as an intermediate in the redox cycling of sulfur in sediments from the oxygen minimum zone from the Arabian Sea. Using metagenomics approach, the authors find the presence of tetrathionate generating, oxidizing or reducing genes and identify bacteria potentially responsible for such processes. Through slurry incubations, the authors show the in-

volvement of tetrathionate in the microbial sulfur cycle in these sediments. Tetrathionate itself was not detected in-situ most, likely due to its reactivity. The authors propose pyrite and/or thiosulfate as potential sources of tetrathionate, which is subsequently oxidized or reduced in the system.

Authors' Response: We thank the Reviewer for his appreciation of the phenomenon unearthed in this study.

Authors' Changes in Manuscript: Not applicable.

Referee's Comment: The sampling approach is rather unusual (subsampling of oxidation critical subsamples from split-cores, see comments below). In addition, the description of the different subsamples are not entirely clear to me, which is most likely a formulation issue (see details below). Description of the analytical methods are not precise (see detailed comments below).

Authors' Response: We agree that there were some deficiencies in our explanation of the sampling and analytical procedures, so in the revised manuscript we have now described these in a more detailed and scientifically explicit manner. Improvements in these regards can be identified through changes in the text shown in the Track Changes file of the manuscript and our responses to the similar points mentioned below in your line-by-line specific comments.

Authors' Changes in Manuscript: In the revised manuscript we have now described the sampling and analytical procedures in a more detailed and scientifically explicit manner.

Referee's Comment: Large parts of the text, especially in the results and discussion part should be rewritten and be more concise. The manuscript contains unnecessary text and phrases, which make reading complicated. Many sentences are too long and sometimes the grammar is not correct such that understanding is in parts not possible. Some examples (but not all) are pointed out/detailed below.

Authors' Response: We have now addressed these concerns by removing the extraneous details, making the language lucid and sentences simple. We believe all these have added to the quality of the writing and comprehensibility of the underlying science.

Authors' Changes in Manuscript: We have now removed the extraneous details, making the language lucid and sentences simple.

Referee's Comment: The figures should be better implemented and explained in the text where appropriate. Downhole analysis of chemical species could be visualized in a depth plot to provide a quicker overview for the reader. The data table can be part of the supplement. Results from the slurry incubations could be presented in an additional figure instead of (or in addition to) the tables. This would help understanding the complex results. From reading it seems very random when and in which samples e.g. tetrathionate is oxidized and at what rates. I strongly suggest splitting results and discussion, this would help to sort out the text and help the reader understanding the story. There is also almost no discussion of the results rather than a presentation, e.g. there is no discussion of the determined rates and what they indicate etc.

Authors' Response: We agree with these suggestions, and have now taken the following measures to ensure that the problems associated with the text and display items are all ameliorated.

Authors' Changes in Manuscript:

- More citations and explanations for the figures have been included in the text. - Analyses of chemical species along the sediment-cores have now been presented in the form of depth plots; accordingly, Table 1 is now moved from the main text to the Supplementary Information. - A new Figure (numbered as 4 in the revised manuscript) has now been incorporated for the results of the slurry incubation experiments. We believe that the visual impact of the data is now instrumental in an easier comprehension of the complex results pertaining to the formation and/or oxidation of tetrathionate, plus the reduction of tetrathionate. - Results and Discussion sections have now been split.

Implications of all the current findings (results/data) have been explained adequately in the new Discussion section.

Referee's Comment: Collectively, I think this manuscript needs a major overhaul with focus on precise description of the sampling and methods and separation of results and methods including a proper and streamlined discussion, before the scientific merit can be judged. The extend of required rewriting including methods, results and discussion extends what is justifiable as a revision. However, I would emphasis a re-submission as a new manuscript once rewritten.

Authors' Response: We have now overhauled the manuscript exactly as suggested.

Authors' Changes in Manuscript: Besides more streamlined description of the methods and sampling procedure, separation of Results and Discussion sections, and result-specific discussions, we have now included new lines of metagenomic, genomic and metatranscriptomic data (and their corresponding discussions) in the revised manuscript. New display items have also been provided for the existing data. Furthermore, we have now deleted a number of such geochemical data that were inexplicable under the current state of knowledge on the biogeochemical system explored in this study.

Referee's Comment: In the following I provide many details, but this may not be complete. Specific comments (incl. few technical comments):

Abstract Line 25: introduce msbl, also: no dash between number and unit (msbl) here.

Authors' Response: We agree, and have now fixed this problem.

Authors' Changes in Manuscript: Mbsl has been defined and the 'dash' removed.

Referee's Comment: Lines 27/28: I suggest to be more precise and name the processes instead of generally speaking about "these metabolisms" or "these processes"

Authors' Response: We agree that the use of precise names for anything technical is

always preferable in scientific literature.

Authors' Changes in Manuscript: However, since in the present case the words "microbial formation, oxidation and reduction of tetrathionate" are appearing twice within the same sentence we prefer to mention the proper nouns initially and use the pronoun "these metabolism" in the second occasion.

Referee's Comment: Line 29: Provide conditions of the incubations under which you could observe tetrathionate generation or turnover. (What types of slurry-incubations, i.e. with amendment of tetrathionate or thiosulfate: : : etc.)

Authors' Response: We agree, and have now provided all the warranted information.

Authors' Changes in Manuscript: Here in the abstract, we have now mentioned the media compositions used in the slurry incubation experiments; more details are already there in the main text.

Referee's Comment: Lines 31/32: Can you calculate a molar concentration or g/sed for iron and manganese instead of giving ppm

Authors' Response: We agree that in any scientific literature molar concentrations are more preferable than ppm values.

Authors' Changes in Manuscript: Incidentally, however, all data and discussions pertaining to the in situ concentrations of iron and manganese have now been removed from the revised manuscript following the suggestions of Reviewer 2.

Referee's Comment: Line 34: instead of "converted" use "oxidized" here and throughout the manuscript (similarity use "reduced" if applicable)

Authors' Response: We agree, now fixed.

Authors' Changes in Manuscript: Instead of "converted" we have used "oxidized" here and throughout the manuscript.

Referee's Comment: Line 35: delete "back"

Authors' Response: Please note that the very preceding sentence reads as "Thiosulfate oxidation by chemolithotrophic bacteria prevalent in situ is the apparent source of tetrathionate in this ecosystem"; so, in this succeeding sentence, we think that it would be more appropriate to write "tetrathionate can be reduced back to thiosulfate" than simply say "tetrathionate can be reduced to thiosulfate".

Authors' Changes in Manuscript: Sentence has been restructured to make the above sense clearer.

Referee's Comment: Line 35: avoid "0" as a concentration, it reads odd. 0 means absence, so 0-2 mM present is wrong as 0 means not present. You could write e.g. up to 2 mM

Authors' Response: We agree.

Authors' Changes in Manuscript: Have now written "up to" wherever applicable.

Introduction

Referee's Comment: Line 45: delete "running"

Authors' Response: We have now done as suggested.

Authors' Changes in Manuscript: Deleted "running".

Referee's Comment: Line 48: delete "So"

Authors' Response: Done.

Authors' Changes in Manuscript: Deleted "so".

Referee's Comment: Line 54: Delete: "In this context" – unnecessary

Authors' Response: We agree.

Authors' Changes in Manuscript: "In this context" is now deleted.

Referee's Comment: Line 55: replace/reformulate "seldom appreciated" by rarely investigated or similar

Authors' Response: We agree.

Authors' Changes in Manuscript: We have now reworded this as suggested.

Material and Methods

Referee's Comment: Line 77: delete "the"

Authors' Response: Here we are referring to two specific cores out of the total 8 collected on board SSK42. Moreover, the first mention of two SSK42 cores constituting the raw material of this study had already happened only seven lines ahead of this line, so the article "the" should be used here ahead of the words "two gravity cores".

Authors' Changes in Manuscript: No change made.

Referee's Comment: Line 78: delete "on which the present study is based" (unnecessary text)

Authors' Response: Here we to introduce the readers to the core nomenclature and at the same time convey that out of the ten SSK42 cores, which are being studied and reported (from distinct biogeochemical perspectives) in a series of recent publications by our group, the 5th and 6th (already named in Fernandes et al., 2018 as SSK42/5 and SSK42/6) are the ones on which the present study is based.

Authors' Changes in Manuscript: We have now restructured this sentence to convey the above sense more appropriately.

Referee's Comment: Lines 81ff: The sampling strategy is unusual. Oxidation sensitive sample were collected after splitting the core into two halves. To prevent oxidation a shower of N2 was applied. How was this realized to ensure that no oxidation occurred? Usually smaller hole round core sections are subsamples inside an anaerobic camber (glove box) or subsamples are taken with cut-off syringes via small holes cut in the side

of a liner or alternative from fresh cuts during sectioning. Al halve split exposes large areas to air even though somehow a N2 shower was installed this sees quite unusual. Was this split done at the entire 3 m core? How was a N2 shower over the 3-m length maintained during the sampling of the 10 – 20 subsamples from each core?

Authors' Response: We have now overhauled the portion dealing with sampling details in such a way that all confusions emanating from the previous text are now resolved. This said, it is noteworthy that answers to these questions were already there in our previous publication Fernandes et al., 2018, which also dealt with these SSK42 cores (albeit form other perspectives) and was cited amply throughout the manuscript, including the sampling-related section. We had therefore thought that further repetition of the details would be unnecessary and also potent causes of unintended self-plagiarism. However, now we understand that as an independent paper this manuscript should carry its own sampling details and have therefore brought back many of those details taking sufficient care of literary repetitiveness.

Authors' Changes in Manuscript: The following has now been added to the revised manuscript.

- In order to protect the ASOMZ sediment-samples from aerial oxidation, the entire cores were not split open into two D-shaped halves directly; instead only one ∼15-cm-long C-shaped part of PVC core-liner was removed at a time, as shown in a new Supplementary Figure added to the revised manuscript. We have now restructured the old sentences and written additional new lines to explain this in an unambiguous way. - The 15 cm length exposed at a time for sampling was constantly and closely showered with high-purity N2 emitted from multiple nozzles fitted to multiple nitrogen-generators. This contrivance was sufficient to prevent atmospheric oxidation of the 15 cm exposed surface of the core. - Immediately after the C-shaped longitudinal part of the PVC core-liner was cut open, top one cm of the exposed surface was scrapped off along the core-circumference using sterile scalpels to eliminate potential contaminations from the core-liners' inner-surfaces and/or sea-waters through which the cores had passed.

- Subsequently, to sample a particular sediment-depth of the core for microbiological studies, an approximately 5-mm-thick sediment-slice (spanning equally on either side of the core-height marking) was scooped out with a sterile scalpel and put into a sterile polypropylene bottle. The head-space of every sample-containing bottle was flushed with high-pure N2, following which it was sealed with Parafilm - At the same time, for on board extraction of pore-waters, sediment-samples from a particular depth were taken out by inserting sterile 50 ml cut off syringes deep inside the core cross-section, multiple times along the circumference on the exposed 'C half'; the samples were immediately collected in sterile 50 ml centrifuge tubes; and all these were carried out under focused streams of high-pure N2.

Referee's Comment: What are the "adequate measures" to avoid contamination? Does this refer only to the use of sterile spatulas? Were the sample bottles autoclaved?

Authors' Response: It was already written in the previous manuscript and has now been made more informative. Also please note that ll the samples vials, spatulas and bottles used for sampling were sterile. They were either autoclaved on board or purchased as sealed packs of gamma irradiation sterilized laboratory wares.

Authors' Changes in Manuscript: The previous sentence (lines 84-86 of previous manuscript) reading "Adequate measures were taken to avoid post-sampling contamination of the native microbial communities and physicochemical alteration of the geochemical properties of the sediments" has now been removed. The previous articulation regarding the measure I question has been edited as "Immediately after the C-shaped longitudinal part of the PVC core-liner was cut open, top one cm of the exposed surface was scraped off along the core-circumference using sterile scalpels to eliminate potential contaminations from the core-liners' inner-surfaces and/or seawaters through which the cores had passed.

Referee's Comment: Line 92: fractions means subsamples?

Authors' Response: Yes, here fractions means subsamples.

Authors' Changes in Manuscript: The term "sample fraction" has now been replaced by "sample replicates".

Referee's Comment: Line 93: "while one fraction, each for chemistry : : :" you mean two subsamples, one for chemistry and one for microbiology?

Authors' Response: The Reviewer's understanding of the sentence is exactly same as what was meant.

Authors' Changes in Manuscript: The term "sample fraction" however has now been replaced by "sample replicates".

Referee's Comment: Line 99: ion chromatography

Author Response: The typing error is now fixed.

Authors' Changes in Manuscript: The typing error is now fixed.

Referee's Comment: Line 101: what is the number in brackets? A catalog number? (should be removed).

Authors' Response: The number in the brackets are model numbers and we think that the IC detector's model number would be important here.

Authors' Changes in Manuscript: However, the column model number has now been deleted.

Referee's Comment: Line 106: "passed through : : :membranes" – the samples were probably "filtered" – with syringe filters?

Authors' Response: Yes, volumes of the samples were already mentioned to be in the order of $\mu$L, so it could be filtered only using a syringe.

Authors' Changes in Manuscript: Following your suggestion we have now added the word "filters" after the word "membrane", so this sentence in the revised manuscript now reads as "pore-water samples were ... passed through 0.22 $\mu$m hydrophilic

polyvinylidene fluoride membrane filters".

Referee's Comment: Line 108: Please more details on the method calibration: What standards were used for calibration, what calibration, how many points? External? What does "sample reproducibility," mean analytical precision? How was this determined, by how many replicate measurements of the same sample? The value should be given in molar concentration if for a specific sample or in RSD (%) if it refers to the precision of the method itself.

Authors' Response: We agree, and have now provided all details for the method of IC calibration.

Authors' Changes in Manuscript: We have now added the following here.

Analytical grade thiosulfate IC Standard [ICS024, Sigma Aldrich, (St. Louis, MO, USA)] was used to prepare the calibration curve for quantification of this anion. Three different concentrations of thiosulfate, 0.5 $\mu$M, 5 $\mu$M and 20 $\mu$M, were measured for the construction of calibration curve by plotting peak height against concentration. Based on triplicate analyses of the standards, deviations from actual concentrations were found to be less than 2.5%.

Referee's Comment: Line 129 ff: for the determination of AVS and CRS fractions, original literature should be cited. How was $Ag_2S$ quantified, gravimetrically?

Authors' Response: Yes, $Ag_2S$ was quantified gravimetrically following Canfield et al., (1986): Canfield, D.E., Raiswell, R., Westrich, J.T., Reaves, C.M., Berner, R.A. 1986. The use of chromium reduction in the analysis of reduced inorganic sulfur in sediments and shales. Chem. Geol. 54, 149-155.

Authors' Changes in Manuscript: However, in agreement with some of the comments of Reviewer 2 we have now removed those portions of the manuscript which envisaged in situ production of tetrathionate/thiosulfate from pyrite; accordingly, methods concerning the estimation of pyrite have also been removed from the revised text.

Referee's Comment: Line 261: 0% partial pressure? Pressure unit is not percent. Also 0 probably means anoxic?

Authors' Response: Yes, 0% partial pressure means anoxic; the workstation was set at this specification following the manufacturer's instruction.

Authors' Changes in Manuscript: Not applicable.

Referee's Comment: Line 268ff: This is the standard cline protocol, which is widely used and generally accepted - not necessary to describe the principle.

Authors' Response: We agree.

Authors' Changes in Manuscript: We have now removed the mention of the principle.

Referee's Comment: Line 286: what does serially diluted mean?

Authors' Response: Serial dilution as a basic microbiological practice is very much similar to that in chemistry. Here a microbial inoculum (cell suspension) is sequentially diluted by orders of magnitude so as to reduce the density of cells in the suspension to such levels which give rise to manually-countable colonies when the suspension is spread out on solid media plates.

Authors' Changes in Manuscript: Not applicable.

Referee's Comment: Line 288: here and elsewhere, please use until instead of till. As this is a scientific article, and till is considered to be informal which should be avoided.

Authors' Response: We agree and have now done so.

Authors' Changes in Manuscript: "Until" used instead of "till".

Referee's Comment: Line 289: "pure-plates"?

Authors' Response: Pure-plates refer to those microbial culture plates which have only one type of colonies, evidently representing one type of bacterium (strain).

Authors' Changes in Manuscript: Not applicable.

Results and discussion

Referee's Comment: Line 327 "relevant (microorganisms)" ?

Authors' Response: The sentence here reads as "Tetrathionate-forming, oxidizing, or respiring genes and relevant microorganisms are abundant in . . .", so it is presumable that the microorganisms, like the genes identified, also refer to tetrathionate forming/oxidizing/reducing entities.

Authors' Changes in Manuscript: Not applicable..

Referee's Comment: Line 329-334: Very long sentence- almost not understandable: consider splitting and rewriting.

Authors' Response: We agree.

Authors' Changes in Manuscript: The entire paragraph spanning between the previous line numbers 329-343 have now been simplified; as a part of this restructuring, the sentence in question has also been split into two.

Referee's Comment: Line 334 "were found to contain" – shorter "contained"

Authors' Response: Done, as suggested.

Authors' Changes in Manuscript: Extra words deleted.

Referee's Comment: Line 341: Unpublished data should not be cited if not necessary. Here are 5 other references given. The reference to unpublished data is unnecessary.

Authors' Response: Done as suggested.

Authors' Changes in Manuscript: The unpublished data are not cited anymore.

Referee's Comment: Line 391-395: Long and unclear sentence.

Authors' Response: We agree.

Authors' Changes in Manuscript: We have now rewritten this sentence in a lucid way.

Referee's Comment: Line 406 – 411: Long and unclear sentence.

Authors' Response: We agree.

Authors' Changes in Manuscript: We have now rewritten this sentence in a lucid way.

Referee's Comment: Line 423: The discussion refers here to another unpublished paper. The suggestion/discussion here is based on unpublished data from the authors. Such data should either be included in this manuscript or published first. Alternatively, the results should be discussed in the light of other already published studies. Otherwise, this discussion is not solid.

Authors' Response: For these data illustrating the feasibility of aerobic metabolism in these sediment horizons, please note that the same constitute a completely separate paper of our group, under consideration elsewhere, and those data are too voluminous to be incorporated here. Anyway, in this paper we have now provided the key data as Supplementary Information so that the reader can make sense of what is there.

Authors' Changes in Manuscript: We have now mentioned in this revised manuscript that genes for aerobic respiration by aa3-/cbb3-type cytochrome-c oxidases (coxABCD / ccoNOPQ) and cytochrome-bd ubiquinol oxidase (cydABX / appX) were identified in the assembled metatranscriptome from 275 cmbsf of SSK42/6 in general, and the portions of the metatranscriptomic dataset in particular which matched with sequences from the tetrathionate-oxidizing isolates, thereby suggesting that potential activity of this aerobic metabolic process is possible in this environment (the relevant gene and transcript Tables have also been incorporated in the revised manuscript as Supplementary Materials).

Referee's Comment: Lines 425 ff. The results of the incubation appear very unsystematic or random. A figure could help for an overview. The writing is also not precise, i.e. the "samples" do not convert thiosulfate to tetrathionate : : : conversion was observed

in the samples or the organisms in the sample convert the species: : :

Authors' Response: We agree.

Authors' Changes in Manuscript: We have now made the presentation of all these results more systematic by adding new Figures and amending the language in the text in such a way that it is clear to the reader that the organisms in the samples converted the sulfur species. Furthermore, we have also amended the corresponding Methods section to put the principles of the slurry incubation experiments and the subsequent data in proper context.

Referee's Comment: Line 430ff: "In contrast, : : :" The sentence is very long. Also, there is no "contrast" obvious. "free and detectable" is unnecessary.

Authors' Response: We agree, but in this context it is also noteworthy that the data presented in the lines 430ff regarding the thiosulfate oxidation property of the 45, 60 and 295 cmbsf communities of SSK42/5 are distinct from those concerning 0, 15, 90 and 160 cmbsf presented just in the previous sentence; hence the succeeding sentence ought to start with "In contrast".

Authors' Changes in Manuscript: We have now simplified this sentence as well as the entire text in this Results section; accordingly, the phrase "free and detectable" has been removed.

Referee's Comment: Lines 434ff: the samples do not metabolize. Organisms have a metabolism but not a sediment samples. In this entire section is not clear how the rates were determined. A figure might help

Authors' Response: We agree.

Authors' Changes in Manuscript: Have now replaced the word "metabolize" by the word "oxidize" in the context of this sentence. In addition, a number of lines within this section have been restructured, while a more elaborate description of the rationale behind the determination of rates has been provided. The entire set of data has also

been presented graphically.

---

## Author Comment (AC8) · 24 Sep 2019

Referee's Comment: The manuscript describes the analysis of inorganic sulfur compound cycling microbial populations in two sediment cores from the Indian Ocean. Particular focus is on populations driving the metabolism of thiosulfate and tetrathionate (thiosulfate reducing/tetrathionate forming, tetrathionate reducing, and tetrathionate oxidising groups). The study used a range of geochemical measurements, slurry incubations, microbiology (isolation of sulfur cycling microorganisms and assessment of their capabilities to transform thiosulfate and tetrathionate) as well as molecular biological approaches (metagenomics and metatranscriptomics). The biogeochemistry of sulfur compounds in sediments is a complex web of chemical and biological transformations, there is a need to better understand the role of individual species of inorganic sulfur compounds as well as the metabolic pathways and microbial groups involved in their transformations. As such, this is a topic of high interest, especially if, as the title implies, some of the transformations may be of a cryptic (not easily identified) nature.

Authors' Response: We thank the Reviewer for appreciating the underlying science of this study.

Authors' Changes in Manuscript: Not applicable.

Referee's Comment: My overarching impression of the manuscript is that it is not easy to follow the story and that it would benefit from revising the structure. It is lacking a clear approach to the analysis and presentation of the data. Even starting in the introduction, I would suggest that, given the focus on the various enzymes being instrumental in the transformations of thiosulfate and tetrathionate, the introduction should provide a brief overview of the most important enzymes involved (and their encoding genes) and perhaps contain a schematic conceptual overview illustrating the most important points.

Authors' Response: We agree with your concerns, and so in the revised version of the manuscript have overhauled the entire text by providing the warranted information (regarding the enzymes and genes which appear to be instrumental in the transformations of thiosulfate and tetrathionate) in the Introduction, making the rationales for the analyses clearer, and discussing the results in their proper context under separate Results / Discussions sections. A schematic conceptual overview illustrating the most important points was already there in the form of Figure 4 in the previous manuscript and the same has now been updated with respect to the altered data components of the paper.

Authors' Changes in Manuscript: As mentioned above.

Referee's Comment: It would be beneficial and aid readability, if a clear overview of the basic findings was shown perhaps as depth profiles showing key chemical parameters of the cores under investigation.

Authors' Response: We agree, and have now done as suggested by the Reviewer.

Authors' Changes in Manuscript: The Results and Discussions sections have now been segregated. We have incorporated new figures showing depth profiles of all the important geochemical and microbiological parameters and comprehensively discussed their implications under a new Discussion section 4.1 titled "Trends of geomicrobial parameters down the sediment-depths corroborated sulfur cycle functions centered on tetrathionate".

Referee's Comment: With a view of the diversity and metagenomics analysis, I have two key criticisms: (i) revolving around the specific use of metagenomics read data for taxonomic assignment and (ii) extrapolating from that assignment to physiological properties of entire genera of bacteria. In that context, I have to say that I think it is a pity the authors did not carry out a diversity analysis of the sediment samples based on pooled 16S rRNA amplicon sequencing in parallel to the metagenomics/metatranscriptomics, because the ribosomal RNA gene survey would provide a much better and more robust diversity analysis than the assignment of taxonomy based on random metagenomic reads. Although a taxonomic assignment of a random metagenomic read is possible, it is fraught with major uncertainty, unless a closely related organism's genome is available in a database. As that is not the case for the vast majority of microorganisms found in nature at present, the taxonomic assignment of metagenomic reads is a bound to provide unreliable/unresolved taxonomies and lead to poor estimates of the abundance of specific types of bacteria. This affects data shown in Tables S8-13 as well as Fig 1 and 2.

Authors' Response: We agree that rRNA gene sequence analyses provide more robust taxonomic diversity analysis than the assignment of taxonomy based on identity

of shotgun metagenomic reads; but on the flip side we have to bear in mind that such data only remain qualitative and give estimates of alpha diversity. We have already published the comprehensive, 16S rRNA gene sequence based, taxonomic diversity analyses for almost 100 sediment-samples from six SSK42 cores, including SSK42/5 and SSK42/6 [Fernandes et al., 2018, Enhanced carbon-sulfur cycling in the sediments of Arabian Sea oxygen minimum zone center. Sci. Rep. 8: 8665]. But such PCR amplified sequence data are not theoretically suitable for interpreting any quantitative population ecology trends or community metabolic functions such as the ones we have revealed in this paper from shotgun metagenomic sequence datasets (by directly annotating the same via BLAST search against the non-redundant protein sequence database, or assembling and annotating genes within the contigs by HMMER search against the EggNOG database, respectively).

We also agree with your concerns that taxonomic assignment from shotgun metagenomic reads, though possible, is fraught with major uncertainty, unless a closely related organism's genome is available in a database. With regard to this issue we can assure you that the parameters we have used to classify reads using the Best Hit Classification algorithm [BlastX search with minimum 45 nucleotides (15 amino acids) alignment and $\geq$60% identity, and maximum e-value allowed 1e–5] are stringent enough to assign taxonomic affiliation to homologs of metabolically diverse genes, irrespective of their intrinsic levels of conservation, in a reliable manner up to the genus level. This stringency level of search parameters is considered optimum across the literature because it neither exaggerates diversity not fails to resolve taxonomies for most categories of genes.

Anyway, these are virtually never-dying debates which yield no clear and unequivocal results, so to lay all apprehensions regarding "taxonomy-physiology disconnect" to rest we have carried out whole genome shotgun sequencing and annotation for the three tetrathionate-forming isolates, the two tetrathionate-oxidizing isolates, and the lone tetrathionate-reducing isolate, and mapped the available metagenomic sequence

data from the 25 distinct sample-sites of SSK42/5 and SSK42/6 separately onto each of above mentioned de novo sequenced genomes – remarkably, significant percentages of the metagenomic read-sets were found in this way to match sequences from the individual genomes. The data, which clearly give a picture of the relative abundances of the strains in each of the 25 distinct sediment-samples have been presented in the form of a new heat map figure.

Authors' Changes in Manuscript: We have now carried out whole genome shotgun sequencing and annotation for the three tetrathionate-forming isolates Halomonas sp. MCC 3301, Methylophaga sp. MTCC 12599 and Pseudomonas bauzanensis MTCC 12600; the two tetrathionate-oxidizing isolates Halothiobacillus sp. SB14A, and Pusillimonas ginsengisoli MTCC12558; and the tetrathionate-reducing isolate Enterobacter sp. RVSM5a. Subsequently we have mapped the metagenomic sequence data from the 25 distinct sample-sites of SSK42/5 and SSK42/6 separately onto each of above mentioned de novo sequenced genomes – remarkably, significant percentages of the metagenomic read-sets were found in this way to match sequences from the individual genomes. The ensuing data, which clearly give a picture of the relative abundances of the strains in each of the 25 distinct sediment-samples, have been presented in the form of a new heat map figure.

Referee's Comment: My second criticism is the assumption in the paper that entire genera of bacteria always share specific physiological capacities with respect to the sulfur transformations of interest. While this is true for some genera (and a parameter used in systematics), for many genera this is not necessarily the case. Therefore, suggesting that genus A or B are tetrathionate producing bacteria or -oxidising bacteria, will likely overestimate the abundance of that specific metabolic type based on that assumption.

Authors' Response: So for as direct taxonomic annotations of raw metagenomic reads, followed by functional/metabolic classification of taxa, are concerned please note that we have all along clarified (already in the initial manuscript) that for one category of genera each and every member strain in the literature is known to possess tetrathionate-forming/oxidizing/reducing property, so the presence of such genera is more definitely indicative of the concerned processes in situ, whereas for another category of genera only some (and not all) member strains are known to possess tetrathionate-forming/oxidizing/reducing property, so their presence indicates further additional possibilities of such processes in situ.

Furthermore, it may be noted that to keep this discrimination explicit, trends of relative abundance for the first category were depicted in Figs. 1 and 2 (these data are clearly free from diversity/abundance over-estimation), while those for the second category were all presented separately in Supplementary Tables S8-S13 (these data are likely to involve unknown proportions of diversity/abundance over-estimation, and so have been kept in isolation from the definitive estimates given in Figs. 1 and 2).

Authors' Changes in Manuscript: We have now edited the text in such a way as to make the above issues more clearly comprehendible.

Referee's Comment: On the other hand, the metagenomics data can reveal the abundance of specific types of genes, as has been done here, and potentially identify the types of bacteria potentially contributing to the cycling of specific compounds. Too much of the discussion of inorganic sulfur metabolism in these sediments is based on the broad assumption of taxonomy, and too little is made of the specific genes found and listed in various supplementary tables. There are still limitations of our understanding of the genetics of sulfur transformations and it would be useful to perhaps illustrate whether the enzymes/genes driving specific transformations of sulfur compounds in some of the taxa mentioned (eg Salmonella) have actually been identified. Very little is done with the metatranscriptome data, it only gets a few mentions, but there is no clear overview of what has been found in which layer, how many reads were analysed and generally which bacteria were transcriptionally active with respect to sulfur cycling.

Authors' Response: We absolutely agree that there are major limitations in global

understanding of the genetics of sulfur transformations; therefore, we have now reviewed in the new Introduction all the enzymes/genes that have been identified thus far as drivers of specific transformations of sulfur compounds in the taxa mentioned/considered in the text (including Salmonella). In this context it is noteworthy that corroborating your assumption, the genome of the current tetrathionate-reducing isolate belonging to the genus Enterobacter was found not to encompass the typical tetrathionate reductase (ttr) genes. For the records, we have now also added a detailed analysis of the six new genomes in relation to the homologs of sulfur-transformation genes present.

Authors' Changes in Manuscript: We have now added a detailed analysis of the six new genomes in relation to the homologs of sulfur-transformation genes present. Subsequently we have mapped the metagenomic sequence data from the 25 distinct sample-sites of SSK42/5 and SSK42/6 separately onto each of above these de novo sequenced genomes. The ensuing data, which clearly give a picture of the relative abundances of the strains in each of the 25 distinct sediment-samples, have been presented in the form of a new heat map figure.

As for the metatranscriptome analyses, we have now added new data identifying which tetrathionate-metabolizing genes in general, and those matching homologs from the genomes of the new isolates in particular, were there in the metatranscriptome. In the revised manuscript we have also included dedicated sections for in depth metatranscriptomic methodology where the all the read statistics have been given. Notably, however, metatranscriptome was analyzed for only 275 cmbsf of SSK42/6, so the data obtained thereof are not applicable to the other layers of the two sediment packages.

Referee's Comment: Regarding the transformations measured in slurry experiments, there needs to be a more complete reporting of the activities measured (or not) in all sediment samples. This should be shown comprehensively, not as currently done in Tables S14-S21, which suggest that only a few subsamples had certain activities. I am also not convinced that a 30-day incubation period of the slurries, some incubating

sediments from a completely anoxic system under aerobic conditions (!), is providing the sort of activity data that would be supportive of suggesting that these key biological reactions are linked up in a cryptic cycle. The mentioning of alternative sources of oxygen from cryptic sources such as perchlorate is pointing to an unpublished study by the same authors. If this is crucial for the understanding of the functioning of this system and the microbiological activities required for the cryptic cycling of these compounds, these aspects should be incorporated here or the other study needs to be published. Alternatively, it is possible that the activities are due to facultative anaerobic bacteria in these sediments that have reverted back to an aerobic lifestyle given suitable incubation conditions and a 30-day period to wake up.

Authors' Response: We agree and have now reported the slurry culture data as graphs plotted against sediment-depths. Notably, whatever may be the actual tetrathionate formation/oxidation rate of the SSK42 sediment-samples in vitro or in situ, results of the slurry culture experiments illustrated that tetrathionate-forming and oxidizing bacteria of SSK42/5 and SSK42/6 were alive in situ. The issue of potential in situ active state of the tetrathionate-metabolizing bacteria was addressed mainly by the metatranscriptomic data, which involved either assembly and general annotation of sequences, or mapping of reads against the tetrathionate-metabolizing genes of the isolates.

As for the data illustrating the feasibility of aerobic metabolism in these sediment horizons, please note that the same constitute a completely separate paper of our group, under consideration elsewhere, and those data are too voluminous to be incorporated here. Anyway, we have now provided that much of data in this revised manuscript which could be sufficient to convey the feasibility of aerobic metabolism in these sediment horizons.

Authors' Changes in Manuscript: We have now added in this revised manuscript that genes for aerobic respiration by aa3-/cbb3-type cytochrome-c oxidases (coxABCD / ccoNOPQ) and cytochrome-bd ubiquinol oxidase (cydABX / appX) were identified in the assembled metatranscriptome from 275 cmbsf of SSK42/6 in general, and the portions of the metatranscriptomic dataset in particular which matched with sequences from the tetrathionate-oxidizing isolates, thereby suggesting that potential activity of this aerobic metabolic process is possible in this environment (the relevant gene and transcript Tables have also been incorporated in the revised manuscript as Supplementary Materials).

Specific comments Referee's Comment: Introduction: The introduction would benefit from a description of relevant metabolic pathways and enzymes targeted by the analysis of this paper Line 73: define mbsl Line 75 what was the diameter of these cores?

Authors' Response: We agree, and have now incorporated the warranted information in the Introduction. The diameter of all SSK42 cores was 12 cm; this information has been incorporated alongside the full-forms of the oceanographic units.

Authors' Changes in Manuscript: As mentioned above.

Referee's Comment: Line 82: for ease of reading I suggest to always refer to both cores with the full abbreviation, not SSK42/5 and 6 but SSK42/5 and SSK42/6

Authors' Response: We agree, and have now done as warranted.

Authors' Changes in Manuscript: Throughout the new manuscript both the cores have been referred to with their full abbreviations, i.e. not as SSK42/5 and 6 but as SSK42/5 and SSK42/6.

Referee's Comment: Line 82: I find the description of the N2 shower lacking in detail and find it hard to understand how it would keep the exposed core adequately protected from oxygen.

Authors' Response: We have now overhauled the portion dealing with sampling details in such a way that all confusions emanating from the previous text are now resolved. This said, it is noteworthy that answers to these questions were already there in our previous publication Fernandes et al., 2018, which also dealt with these SSK42 cores (albeit form other perspectives) and was cited amply throughout the manuscript, including the sampling-related section. We had therefore thought that further repetition of the details would be unnecessary and also potent causes of unintended self-plagiarism. However, now we understand that as an independent paper this manuscript should carry its own sampling details and have therefore brought back many of those details taking sufficient care of literary repetitiveness.

Authors' Changes in Manuscript: The following has now been added to the revised manuscript.

- In order to protect the ASOMZ sediment-samples from aerial oxidation, the entire cores were not split open into two D-shaped halves directly; instead only one ∼15-cm-long C-shaped part of PVC core-liner was removed at a time, as shown in a new Supplementary Figure added to the revised manuscript. We have now restructured the old sentences and written additional new lines to explain this in an unambiguous way.

- The 15 cm length exposed at a time for sampling was constantly and closely showered with high-purity N2 emitted from multiple nozzles fitted to multiple nitrogen-generators. This contrivance was sufficient to prevent atmospheric oxidation of the 15 cm exposed surface of the core.

- Immediately after the C-shaped longitudinal part of the PVC core-liner was cut open, top one cm of the exposed surface was scrapped off along the core-circumference using sterile scalpels to eliminate potential contaminations from the core-liners' inner-surfaces and/or sea-waters through which the cores had passed.

- Subsequently, to sample a particular sediment-depth of the core for microbiological studies, an approximately 5-mm-thick sediment-slice (spanning equally on either side of the core-height marking) was scooped out with a sterile scalpel and put into a sterile polypropylene bottle. The head-space of every sample-containing bottle was flushed with high-pure N2, following which it was sealed with Parafilm.

- At the same time, for on board extraction of pore-waters, sediment-samples from a

particular depth were taken out by inserting sterile 50 ml cut off syringes deep inside the core cross-section, multiple times along the circumference on the exposed 'C half'; the samples were immediately collected in sterile 50 ml centrifuge tubes; and all these were carried out under focused streams of high-pure N2.

Referee's Comment: Line 88: scraped

Authors' Response: We apologise, the typo is now fixed.

Authors' Changes in Manuscript: The spelling mistake has been rectified.

Referee's Comment: Line 145 define cmbsf, how thick was that sediment layer?

Authors' Response: cmbsf is now defined; approximately 5-mm-thick sediment-slices were sampled.

Authors' Changes in Manuscript: The above points have now been incorporated in the revised manuscript.

Referee's Comment: Line 147 and 190: pooling up, no 'up'

Authors' Response: We agree, now fixed.

Authors' Changes in Manuscript: The word 'up' has now been deleted.

Referee's Comment: Line 213: kmer lenghts of . . .

Authors' Response: Thank you, "lengths of" now added.

Authors' Changes in Manuscript: As mentioned above.

Referee's Comment: Line 227: text string search?

Authors' Response: Yes.

Authors' Changes in Manuscript: Not applicable.

Referee's Comment: Line 236/7: not every read would represent one of these functional groups. Please reword/revise or explain more clearly what you did. There are no 'genera' of this that and the other, they are genera that contain species some of which have certain physiological characteristics, but not necessarily all of them

Authors' Response: We agree that your concerns hold well in a large number of cases, and exactly for that reason, we had clearly distinguished (already in the initial manuscript) that for one category of genera each and every member strain in the literature is known to possess tetrathionate-forming/oxidizing/reducing property, so the presence of such genera is more definitely indicative of the concerned processes in situ, whereas for another category of genera only some (and not all) member strains are known to possess tetrathionate-forming/oxidizing/reducing property, so their presence indicates further additional possibilities of such processes in situ.

Furthermore, it may be noted that to keep this discrimination explicit, trends of relative abundance for the first category were depicted in Figs. 1 and 2 (these data are clearly free from diversity/abundance over-estimation), while those for the second category were all presented separately in Supplementary Tables S8-S13 (these data are likely to involve unknown proportions of diversity/abundance over-estimation, and so have been kept separate from the definitive estimates given in Figs. 1 and 2).

Authors' Changes in Manuscript: We have now edited the text in such a way as to make these issues more clearly comprehendible.

Referee's Comment: Line 246 following: No context for why one would assess the aerobic metabolism of these compounds with samples from 275cm below the sediment surface where there is no oxygen.

Authors' Response: Please note that since this study was aimed at revealing potential roles of tetrathionate in the sulfur cycle of marine sediments, microbial communities present at all individual sediment-depths (from surface to core-bottom) of the two cores explored (and not just 275 cmbsf alone) were tested for their potential abilities to oxidize thiosulfate to tertahionate, and oxidize/reduce tetrathionate. Pure culture isolations and metatranscriptome analysis, however, were done only from 275 cmbsf, which is within the sulphate-methane transition zone where microbiological activity, for any marine sediment horizon, is generally very high and multi-faceted. Authors' Changes in Manuscript: We have now restructured this sentence to convey the above sense more appropriately.

Referee's Comment: Line 309: actually described below

Authors' Response: We apologise.

Authors' Changes in Manuscript: The oversight is now corrected.

Referee's Comment: Line 327: reword, genes do not have these activities, they encode enzymes that transform the compounds

Authors' Response: We agree, now fixed.

Authors' Changes in Manuscript: The sentence in previous Line 327, plus those in other similar contexts, has now been reworded so as to give the right biological sense.

Referee's Comment: Line 350: I do not think that all of marine Pseudomonas and Halomonas do that

Authors' Response: We agree that not all but majority of the marine strains of Pseudomonas and Halomonas have the metabolic capacity of conversion of thiosulfate into tetrathionate. Anyway, we have now overcome this small but important question mark by carrying out whole genome shotgun sequencing and annotation for the current three tetrathionate-forming isolates Halomonas sp. MCC 3301, Methylophaga sp. MTCC 12599 and Pseudomonas bauzanensis MTCC 12600, subsequent to which we have mapped the metagenomic sequence data from the 25 distinct sample-sites of SSK42/5 and SSK42/6 separately onto each of above mentioned de novo sequenced genomes – remarkably, significant percentages of the metagenomic read-sets were found in this way to match sequences from the individual genomes. The data, which clearly give a picture of the relative abundances of the strains in each of the 25 distinct sedimentsamples, have been presented in the form of a new heat map figure.

Authors' Changes in Manuscript: We have now carried out whole genome shotgun sequencing and annotation for the current three tetrathionate-forming isolates Halomonas sp. MCC 3301, Methylophaga sp. MTCC 12599 and Pseudomonas bauzanensis MTCC 12600, subsequent to which we have mapped the metagenomic sequence data from the 25 distinct sample-sites of SSK42/5 and SSK42/6 separately onto each of above mentioned de novo sequenced genomes. The data, which clearly give a picture of the relative abundances of the strains in each of the 25 distinct sediment-samples, have been presented in the form of a new heat map figure.

Referee's Comment: Table S1- Please clarify what is meant with 1st or 2nd sample fraction

Authors' Response: To sample a particular sediment-depth of a core for microbiological studies, an approximately 5-mm-thick sediment-slice (spanning equally on either side of the core-height marking) was scooped out with a sterile scalpel and put into a sterile polypropylene bottle. For every sediment-depth, two such sample-replicates or slices - designated for duplicate metagenome (plus other metaomics) analyses - were collected (these were designated as sample replicates 1 and 2); a third slice was taken for all culture-dependent studies (notably, in the revised manuscript, the word "sample fraction" is now replaced by "sample replicate").

Authors' Changes in Manuscript: The warranted clarification has now been added to the revised text, as stated above.

Referee's Comment: Table S4- Not a single DoxA encoding gene was identified in any of the samples. Should it perhaps not be listed in Table S4 accordingly? Please state what the significance of the yellow highlighting is in this and the other excel based supplementary tables

Authors' Response: Thanks for the suggestions. We have now removed the mention

**BGD**

of doxA and soxD from Tables S3 and S4.

As the supplementary tables are large and each excel sheet contains large number of genes we have tried to assist visual cognition by alternately highlighting genes with similar metabolic function.

Authors' Changes in Manuscript: We have now removed the mention of doxA and soxD from Tables S3 and S4.

Referee's Comment: Table S6 Define what is meant with prevalence and how it is quantified Unclear what is meant with correlation of metabolic type with sediment depth, when depth is not quantified here

Authors' Response: To reduce the complexity of some of the sentences, the term "prevalence" has been used to denote relative abundance (detailed description for the estimation of relative abundance from metagenomic data, and its synonymy with prevalence, were mentioned in the Methods).

Please note that both the ∼3-m-long cores were explored biogeochemically at 15 to 30 cm intervals, which have been referred to here as each sediment-depth. In this way, we considered sediment-depth and tetrathionate metabolizing bacterial groups as two quantitative parameters and tested their interdependence by determining Pearson correlation coefficient (CC) and/or Spearman rank correlation coefficient (RCC).

Authors' Changes in Manuscript: The detailed method for the estimation of relative abundance of taxa via direct taxonomic annotation of raw metagenomic reads, and the synonymy between "relative abundance" and "prevalence", have now been explained in the text in a clearer way.

Referee's Comment: Tables S8 to S11 should have totals for the abundance of all types per depth.

Authors' Response: We agree.

Authors' Changes in Manuscript: We have now given these totals.